# VICON: Vision In-Context Operator Networks for Multi-Physics Fluid Dynamics Prediction

## Abstract

In-Context Operator Networks (ICONs) have demonstrated the ability to learn operators across diverse partial differential equations using few-shot, in-context learning. However, existing ICONs process each spatial point as an individual token, severely limiting computational efficiency when handling dense data in higher spatial dimensions. We propose *Vision In-Context Operator Networks* (VICON), which integrates vision transformer architectures to efficiently process 2D data through patch-wise operations while preserving ICON's adaptability to multiphysics systems and varying timesteps. Evaluated across three fluid dynamics benchmarks, VICON significantly outperforms state-of-the-art baselines: DPOT and MPP, reducing the averaged last-step rollout error by 37.9% compared to DPOT and 44.7% compared to MPP, while requiring only 72.5% and 34.8% of their respective inference times. VICON naturally supports flexible rollout strategies with varying timestep strides, enabling immediate deployment in *imperfect measurement systems* where sampling frequencies may differ or frames might be dropped—common challenges in real-world settings—without requiring retraining or interpolation. In these realistic scenarios, VICON exhibits remarkable robustness, experiencing only 24.41% relative performance degradation compared to 71.37%-74.49% degradation in baseline methods, demonstrating its versatility for depolying in realistic applications.

## 1 Introduction

Machine learning has emerged as a powerful tool for solving Partial Differential Equations (PDEs). Traditional approaches primarily operate on discrete representations, with Convolutional Neural Networks (CNNs) excelling at processing regular grids [1], Graph Neural Networks (GNNs) handling unstructured meshes [2, 3, 4], and transformers capturing dependencies between sampling points within the same domains [5, 6, 7]. However, these discrete approaches face a fundamental limitation: they lack PDE context information (such as governing parameters) and consequently cannot generalize to new parameters or PDEs beyond their training set.

This limitation led to the development of "operator learning," where neural networks learn mappings from input functions to output functions, such as from initial/boundary conditions to PDE solutions. This capability was first demonstrated using shallow neural networks [8, 9], followed by specialized architectures including the widely-adopted Deep Operator Network (DeepONet) [10], Fourier Neural Operator (FNO) [11, 12], and recent transformer-based operator learners [13, 14, 15]. While these methods excel at learning individual parametric PDEs, they cannot generalize across different PDE types, necessitating costly retraining for new equations or even different time step sizes.

Inspired by the success of large language models in multi-domain generation, researchers have explored "Multi-Physics PDE models" to address this single-operator limitation. Common approaches

employ pre-training and fine-tuning strategies [16, 17, 18], though these require substantial additional data (typically hundreds of frames) and fine-tuning before deployment. This requirement severely limits their practical applications in scenarios like online control and data assimilation, where rapid adaptation with limited data sampling is essential.

To eliminate the need for additional data collection and fine-tuning, several works have attempted to achieve few-shot or zero-shot generalization across multi-physics systems by incorporating explicit PDE information. For example, the PROSE approach [19, 20, 21, 22] processes the symbolic form of governing equations using an additional transformer branch. Similarly, PDEformer [23] converts PDEs into directed graphs as input for graph transformers, while others embed symbol-tokens directly into the model [24]. Despite these advances, such methods require explicit knowledge of the underlying PDEs, which is often unavailable when deploying models in new environments.

To further lower barriers for applying multi-physics models in real-world applications, the In-Context Operator Network (ICON) [25] offers a fundamentally different approach: ICON implicitly encodes system dynamics through a few input-output function pairs, then extracts dynamics from these pairs in an in-context fashion. This approach enables few-shot generalization without retraining or explicit PDE knowledge, while the minimal required input-output pairs can be readily collected during deployment. A single ICON model has demonstrated success in handling both forward and inverse problems across various ODEs, PDEs, and mean field control scenarios.

However, ICON's computational efficiency becomes a critical bottleneck when handling dense data in higher dimensions. Specifically, ICON treats each sampled spatial point as an individual token, leading to quadratic computational complexity with respect to the number of points. This makes it computationally prohibitive for practical 2D and 3D applications—to date, only one instance of a 2D case using sparse data points has been demonstrated [25].

To address this limitation, we propose *Vision In-Context Operator Networks* (VICON), which leverage vision transformers to process 2D functions using an efficient patch-wise approach. Going beyond classic vision transformers that typically process patches from a single image, VICON extends the architecture to handle sequences of input-output function pairs, enabling "next function prediction" capabilities while preserving the benefits of in-context operator learning.

Our contributions include:

- Development of VICON, a vision transformer-based in-context operator network for two-dimensional time-dependent PDEs that maintains the flexibility of in-context operator learning while efficiently handling dense data in higher dimensions.

- Comprehensive evaluation across diverse fluid dynamics systems comprising 879K frames with varying timestep sizes, demonstrating substantial improvements over state-of-the-art models MPP [26] and DPOT [27] in both accuracy and efficiency. Our approach reduces the averaged last-step rollout error by 37.9% compared to DPOT and 44.7% compared to MPP, while requiring only 72.5% and 34.8% of their respective inference times.

- Enhanced flexibility supporting varying timestep strides and non-sequential measurements, offering substantial robustness in realistic scenarios with imperfect data collection, e.g. with missing frames. Under these scenarios, VICON exhibits only 24.41% relative performance degradation versus 71.37%-74.49% degradation in baseline methods.

## 2 Related Works

**Operator Learning.** Operator learning [8, 9, 11, 10] addresses the challenge of approximating operators $G : U \rightarrow V$, where $U$ and $V$ are function spaces representing physical systems and differential equations, e.g., $G$ maps initial/boundary conditions or system parameters to corresponding solutions. Among the various operator learning approaches, DeepONets [10, 28] employ branch and trunk networks to independently process inputs and query points, while FNOs [11] leverage fast Fourier transforms to efficiently compute kernel integrations for PDE solutions in regular domains. These methods have been extended to incorporate equation information [29, 30], multiscale features [31, 32, 33], adaptation to heterogeneous and irregular meshes [34, 35, 36, 37].

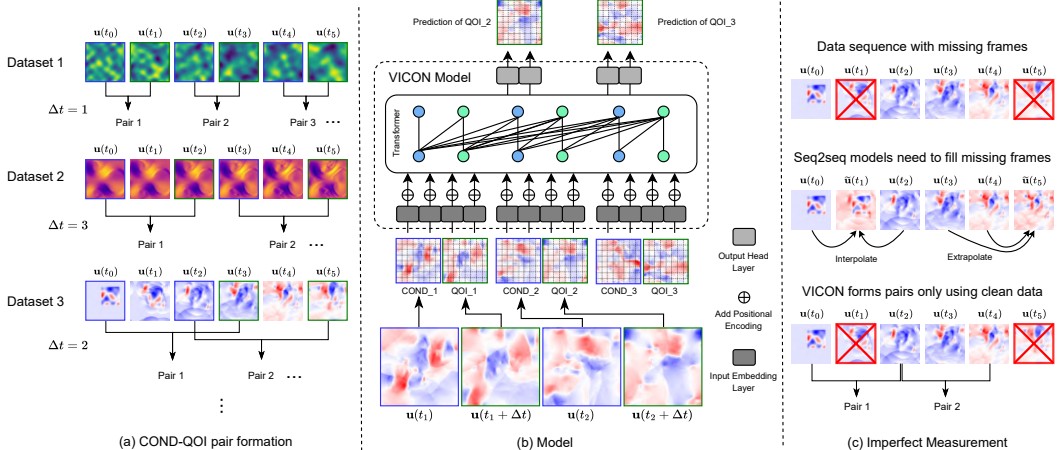

Figure 1: **VICON model overview.** (a) The formation process for conditions (COND) and quantities of interest (QOI) pairs. $\Delta t$ is randomly sampled during training. (b) Model illustration. The inputs to the model are pairs of COND and QOI, which are patchified and flattened before feeding into the transformer layers. The outputs, which represent different patches in the output frame, are transformed back to obtain the final predictions. (c) With imperfect temporal measurements, VICON forms pairs using only clean data, and does not need to fill missing frames.

However, these approaches typically learn a single operator, requiring retraining when encountering different PDE types or time-step sizes. Our work aims to develop a unified model for multiple PDEs with few-shot generalization capabilities based on limited observation frames.

**Multi-Physics PDE Models.** Foundation models in natural language processing [38, 39] and computer vision [40] have demonstrated remarkable versatility across diverse tasks. Drawing inspiration from this paradigm, recent research has explored developing unified models for multiple PDEs in domains such as PDE discovery [41] and computational fluid dynamics [42]. Several approaches follow pre-training and fine-tuning strategies [16, 17], though these require additional resources for data acquisition and fine-tuning before deployment. In parallel, researchers have developed zero- or few-shot multi-physics models by incorporating additional information; for instance, the PROSE approach [19, 20, 21, 22, 43] directly encodes symbolic information of PDEs into the model. Other methods achieving zero- or few-shot generalization include using physics-informed tokens [24], representing PDE structures as graphs [23], or conditioning transformers on PDE descriptions [44]; however, all these techniques require prior knowledge of the PDEs as additional inputs. We select two state-of-the-art models, MPP [26] and DPOT [27], as our baselines.

**In-Context Operator Networks.** ICONs [25, 45, 46] learn operators by observing input-output function pairs, enabling few-shot generalization across various PDEs without requiring explicit PDE representations or fine-tuning. These networks have demonstrated success in handling forward and inverse problems for ODEs, PDEs, and mean-field control scenarios. However, ICONs face significant computational challenges when processing dense data, as they represent functions through scattered point tokens, resulting in quadratic computational complexity with respect to the number of sample points. This limitation has largely restricted their application to 1D problems, with only sparse sampling feasible in 2D cases [25]. Our work addresses these limitations by introducing a vision transformer architecture that efficiently handles dense 2D data while preserving the benefits of in-context operator learning.

# 3 Preliminaries

ICONs were introduced and developed in [25, 45, 46]. They adapt the in-context learning framework from large language models, aiming to train a model that can learn operators from prompted function examples.

Denote an ICON model as $\mathcal{T}_\theta$, where $\mathcal{T}$ is a transformer with trainable parameters $\theta$. The model takes input as a sequence comprising $I$ pairs of conditions ($c$) and quantities of interest ($q$) as $\{\langle c_i, q_i \rangle\}_{i=1}^I$, where each $c$, $q$ contains multiple tokens representing a single function. The model outputs tokens representing future functions at the positions of $c$, i.e. "next function prediction" similar to "next token predictions" in LLM. More precisely, for $J \in \{1, \ldots, I-1\}$, the model predicts $\tilde{q}_{J+1}$ given the leading $J$ pairs and the condition $c_{J+1}$:

$$\tilde{q}_{J+1} = \mathcal{T}_\theta[c_{J+1}; \{\langle c_i, q_i \rangle\}_{i=1}^J]. \tag{1}$$

For training parallelization, ICON uses a special causal attention mask to perform autoregressive learning (i.e., the model only sees the leading $J$ pairs and $c_{J+1}$ when predicting $\tilde{q}_{J+1}$) and enables output of all predictions within a single forward pass [45]:

$$\{\tilde{q}_i\}_{i=1}^I = \mathcal{T}_\theta[\{\langle c_i, q_i \rangle\}_{i=1}^I]. \tag{2}$$

Training loss is computed as the mean squared error (MSE) between the predicted states $\tilde{q}_i$ and the ground truth states $q_i$. To ensure that the model receives sufficient contextual information about the underlying dynamics, we only compute errors for the indices $i > I_{\min}$, effectively requiring at least $I_{\min}$ examples in context. This approach has shown empirical improvements in model performance.

After training, ICON can process a new set of $\langle c, q \rangle$ pairs in the forward pass to in-contextually learn operators and employ them to predict future functions using new conditions:

$$\tilde{q}_k = \mathcal{T}_\theta[c_k; \{\langle c_i, q_i \rangle\}_{i=1}^J], \tag{3}$$

where $J \in \{I_{\min}, \ldots, I-1\}$ is the number of in-context examples, $I_{\min}$ is the number of context exempted from loss calculation, and $k > I$ is the index of the condition for future prediction.

Importantly, all pairs in the same sequence must be formed with the same operator mapping (i.e., the same PDE and consistent timestep size), while operators can vary across different sequences.

# 4 Methodology

## 4.1 Problem Setup

We consider the forward problem for multiple time-dependent PDEs that are *temporally homogeneous Markovian*, defined on domain $\Omega \subseteq \mathbb{R}^2$ with solutions represented by $\mathbf{u}(x,t): \Omega \times [0,T] \to \mathbb{R}^c$, where $c$ is the number of channels. Given the initial $I_0$ frames, $\{\mathbf{u}_i = \mathbf{u}(\cdot, t_i) \mid i = 1, \ldots, I_0\}$, the task is to predict future solutions.

Each solution frame is discretized as a three-dimensional tensor $\mathbf{u}_t = \mathbf{u}(\cdot, t) \in \mathbb{R}^{N_x \times N_y \times c}$, where $N_x$ and $N_y$ are the spatial grid sizes. Due to the homogeneous Markovian property, given the initial data $\{\mathbf{u}_t \mid t \geq 0\}$ from some PDE indexed by $i_p$, for a fixed $\Delta t$, there exists an operator $\mathcal{L} = \mathcal{L}_{\Delta t}^{(i_p)}$ that maps frame $\mathbf{u}_t$ to frame $\mathbf{u}_{t+\Delta t}$ for any $t \geq 0$. Our goal is to train a model that learns these operators $\mathcal{L}_{\Delta t}^{(i_p)}$ from the sequence of function pairs generated from the initial frames. Notably, even within the same dataset and fixed $\Delta t$, different trajectories can exhibit different dynamics (e.g., due to different Reynolds numbers $Re$). These variations are implicitly captured by the function pairs in our framework.

## 4.2 Vision In-Context Operator Networks

The original ICONs represent functions as scattered sample data points, where each data point is projected as a single token. For higher spatial dimensions, this approach necessitates an excessively large number of sample points and extremely long sequences in the transformer. Due to the inherent quadratic complexity of the transformer, this approach becomes computationally infeasible for high-dimensional problems.

Inspired by the Vision Transformer (ViT) [47], we address these limitations by dividing the physical fields into patches, where each patch is flattened and projected as a token. This approach, which we call Vision In-Context Operator Networks (VICON), has its forward process illustrated in Figure 1(b) and detailed in the following.

First, the input $\boldsymbol{u}_t$ and output $\boldsymbol{u}_{t+\Delta t}$ functions are divided into patches $\{\boldsymbol{C}^k \in \mathbb{R}^{R_x \times R_y \times c}\}_{k=1}^{N_c}$ and $\{\boldsymbol{Q}^l \in \mathbb{R}^{R_x \times R_y \times c}\}_{l=1}^{N_q}$, where $R_x, R_y$ are the resolution dimensions of the patch and $N_c$ and $N_q$ are the number of patches for the input and output functions, respectively. While $N_q$ can differ from $N_c$ (for instance, when input functions require additional boundary padding, resulting in $N_c > N_q$), we maintain $N_c = N_q$ throughout our experiments. For notational clarity, we add the subscript $i \in \{1, \ldots, I\}$ to denote the pair index, where $\boldsymbol{C}_i^k$ represents the $k$-th patch of the input function and $\boldsymbol{Q}_i^l$ represents the $l$-th patch of the output function in the $i$-th pair. These patches are then projected into a unified $d$-dimensional latent embedding space using a shared learnable linear function $f_\phi : R_x \times R_y \times c \to \mathbb{R}^d$:

$$\hat{\boldsymbol{c}}_i^k = f_\phi(\boldsymbol{C}_i^k), \quad \hat{\boldsymbol{q}}_i^l = f_\phi(\boldsymbol{Q}_i^l), \tag{4}$$

where $k = 1, \ldots, N_c$ and $l = 1, \ldots, N_q$ are the index of patches.

We inject two types of learnable positional encoding before feeding the embeddings into the transformer: (1) patch positional encodings to indicate relative patch positions inside the whole domain, denoted as $\mathbf{E}_p \in \mathbb{R}^{N_p \times d}$, where $N_p = \max\{N_c, N_q\}$; (2) function positional encodings to indicate whether a patch belongs to a input function (using $\mathbf{E}_c \in \mathbb{R}^{I \times d}$) or output function (using $\mathbf{E}_q \in \mathbb{R}^{I \times d}$), as well as their indices in the sequence:

$$\boldsymbol{c}_i^k = \hat{\boldsymbol{c}}_i^k + \mathbf{E}_p(k) + \mathbf{E}_c(i), \quad \boldsymbol{q}_i^l = \hat{\boldsymbol{q}}_i^l + \mathbf{E}_p(l) + \mathbf{E}_q(i). \tag{5}$$

The embeddings $\boldsymbol{c}_i^k$ and $\boldsymbol{q}_i^l$ are then concatenated to form the input sequence for the transformer:

$$\mathbf{c}_1^1 \ldots \mathbf{c}_1^{N_c}, \mathbf{q}_1^1 \ldots \mathbf{q}_1^{N_q}, \ldots, \mathbf{c}_I^1 \ldots \mathbf{c}_I^{N_c}, \mathbf{q}_I^1 \ldots \mathbf{q}_I^{N_q}.$$

To support autoregressive prediction similar to Equation (2), VICON employs an alternating-sized (in the case where $N_c \neq N_q$) block causal attention mask, as opposed to the conventional triangular causal attention mask as in mainstream generative large language models [38] and large vision models [48]. The mask is defined as follows:

$$\mathbf{M} = \begin{bmatrix} \mathbf{1}_{N_c,N_c} & \mathbf{0} & \cdots & \mathbf{0} & \mathbf{0} \\ \mathbf{1}_{N_q,N_c} & \mathbf{1}_{N_q,N_q} & \cdots & \mathbf{0} & \mathbf{0} \\ \vdots & \vdots & \ddots & \vdots & \vdots \\ \mathbf{1}_{N_c,N_c} & \mathbf{1}_{N_c,N_q} & \cdots & \mathbf{1}_{N_c,N_c} & \mathbf{0} \\ \mathbf{1}_{N_q,N_c} & \mathbf{1}_{N_q,N_q} & \cdots & \mathbf{1}_{N_q,N_c} & \mathbf{1}_{N_q,N_q} \end{bmatrix} \tag{6}$$

where $\mathbf{1}_{m \times n}$ denotes an all-ones matrix of dimension $m \times n$, and $\mathbf{0}$ represents a zero matrix of the corresponding dimensions.

After obtaining the output tokens from Equation (2), we extract the tokens corresponding to the input patch indices $\mathbf{c}_i^1, \ldots, \mathbf{c}_i^{N_c}$ and denote them as $\tilde{\boldsymbol{q}}_i^1, \ldots, \tilde{\boldsymbol{q}}_i^{N_c}$. These tokens are then projected back to the original physical space using a shared learnable linear function $g_\psi : \mathbb{R}^d \to \mathbb{R}^{R_x \times R_y \times c}$:

$$\tilde{\boldsymbol{Q}}_i^l = g_\psi(\tilde{\boldsymbol{q}}_i^l), \tag{7}$$

which predicts $\boldsymbol{Q}_i^l$.

### 4.3 Prompt Normalization

To address the varying scales across different channels and prompts (i.e. sequences of pairs $\{\langle \boldsymbol{c}_i, \boldsymbol{q}_i \rangle\}_{i=1}^I$), we normalize the data before feeding them into the model. A crucial requirement is to consistently normalize functions within the same sequence, to ensure that the same operator is learned. Denoting the normalization operators for $\boldsymbol{c}$ and $\boldsymbol{q}$ as $\mathcal{N}_c$ and $\mathcal{N}_q$ respectively, and the operator in the original space as $\mathcal{L}$, the operator in the normalized space $\mathcal{L}'$ follows:

$$\mathcal{L}'(\mathcal{N}_c(\mathbf{u})) = \mathcal{N}_q(\mathcal{L}(\mathbf{u})). \tag{8}$$

In this work, we simply set $\mathcal{N}_c = \mathcal{N}_q$, mainly because our maximum timestep stride $s_{\max} = 5$ is relatively small (i.e., the scale distribution does not change dramatically). Specifically, we computed the channel-wise mean $\mu$ and standard deviation $\sigma$ of $\{\boldsymbol{c}_i\}_{i=1}^I$ in each prompt, then used these values to normalize $\{\langle \boldsymbol{c}_i, \boldsymbol{q}_i \rangle\}_{i=1}^I$ in that prompt. To avoid division by zero, we set a minimum threshold of $10^{-4}$ for the standard deviation.

## 4.4 Datasets and Data Augmentation

We evaluate on three fluid dynamics datasets representing different physical regimes: 1) PDEArena-Incomp [1] (incompressible Navier-Stokes equations), containing 2,496/608/608 trajectories (train/valid/test) with 56 timesteps each; 2) PDEBench-Comp-HighVis [49] (compressible Navier-Stokes with high viscosity), containing 40,000 trajectories of 21 timesteps each; and 3) PDEBench-Comp-LowVis [49] (compressible Navier-Stokes with numerically zero viscosity), containing 4,000 trajectories of 21 timesteps each. For PDEBench-Comp-HighVis and PDEBench-Comp-LowVis, we randomly split trajectories in 80%/10%/10% proportions for training/validation/testing, respectively.

The temporally homogeneous Markovian property (Section 4.1) enables natural data augmentation for training and flexible rollout strategy for inference (Section 4.5): by striding with larger timesteps to reduce the number of autoregressive steps in long-term prediction tasks: $\Delta t = s\Delta\tau_{i_p} \mid s = 1, 2, \ldots, s_{\max}$, where $\tau_{i_p}$ is the timestep size for recording in trajectory $i_p$. During training, for each trajectory in the training set, we first sample a stride size $s \sim \mathcal{U}\{1, \ldots, s_{\max}\}$, then form $\langle \mathbf{u}_t, \mathbf{u}_{t+s\Delta\tau} \rangle$ pairs for the corresponding operator. We illustrate this augmentation process in Figure 1(a), where different strides are randomly sampled during dataloading.

Additional dataset details appear in Appendix A.

## 4.5 Inference with Flexible Strategies

As mentioned in Section 4.4, VICON can make predictions with varying timestep strides. Given $I_0$ initial frames $\{\mathbf{u}_i\}_{i=1}^{I_0}$, we can form in-context example pairs $\{\langle \mathbf{u}_i, \mathbf{u}_{i+s} \rangle\}_{i=1}^{I_0-s}$. For $j \geq 1$, we set the question condition ($\mathbf{c}_k$ in Equation (3)) as $\mathbf{u}_j$ to predict $\mathbf{u}_{j+s}$, enabling $s$-step prediction.

For long-term rollout, we employ VICON in an autoregressive fashion. The ability to predict with different timestep strides enables various rollout strategies. We explore two natural approaches: single-step rollout and flexible-step rollout, where the latter advances in larger strides to reduce the number of rollout steps.

For single-step rollout, we simply follow the autoregressive procedure with $s = 1$. Flexible-step rollout involves a more sophisticated approach: given a maximum prediction stride $s_{\max}$, we first make sequential predictions with a gradually growing $s = 1$ to $s_{\max}$ using $\mathbf{u}_{I_0}$ as the question condition, obtaining $\{\mathbf{u}_{I_0+s}\}_{s=1}^{s_{\max}}$. Using each frame in this sequence as a question condition, we then make consistent $s_{\max}$-step predictions while preserving all intermediate frames. This flexible-step strategy follows the approach in [46].

This strategy is also applicable to the imperfect measuring cases where partial input frames are missing, eg, due to sensor device error. In this case, we can still form pairs from the remaining frames, with minor adjustment to the strategy generation algorithms, as shown in Figure 1(c).

More details (including algorithms and examples) on rollout strategies are provided in Appendix C.1 (for perfect measurements) and in Appendix C.2 (for imperfect measurements).

## 4.6 Evaluation Metric

We evaluate the rollout accuracy of VICON using two different strategies described in Section 4.5. The evaluation uses relative and absolute $L^2$ errors between the predicted and ground truth frames, starting from the frame $I_0 + 1$. For relative scaling coefficients, we use channel-wise scaling standard deviation $\sigma$ of ground truth frames, which vary between different prompts.

## 5 Experimental Results

We benchmark VICON against two state-of-the-art sequence-to-sequence models: DPOT [27] (122M parameters) and MPP [26] (AViT-B architecture, 116M parameters). The vanilla ICON, which would require processing over 114K tokens per frame (128×128×7), exceeds our available GPU memory and is thus computationally infeasible for direct comparison on these datasets.

For evaluation, we use both absolute and relative $L^2$ RMSE (Section 4.6) on rollout predictions. Since VICON offers the flexibility of predictions with different time step strides, we evaluate our

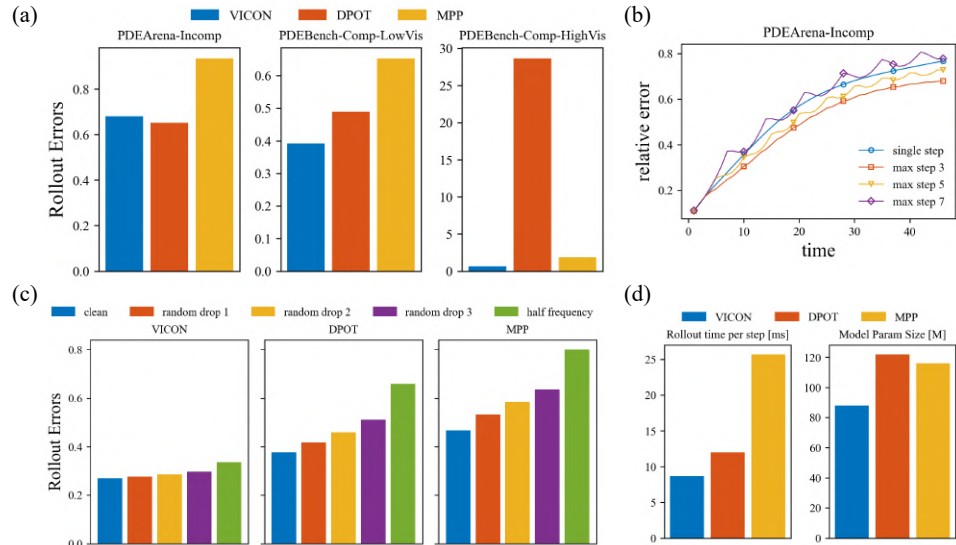

Figure 2: **Main experiment results.** (a) Last step rollout errors on 3 datasets. VICON outperforms MPP on all datasets and outperforms DPOT on 2 datasets. (b) VICON allows flexible rollout strategies to reduce error accumulation and demonstrates stride extrapolation. (c) VICON is robust to imperfect temporal measurements, while MPP and DPOT suffer from performance degradation. (d) VICON is smaller in size and has faster rollout time per step.

single trained model using both single-step and flexible-step rollout strategies (Section 4.5) during inference.

For conciseness, we present summarized plots and tables the main text, and defer complete results and visualizations to Appendix D. Ablation studies examining key design choices of VICON—including patch resolution, positional encoding, and context length—are presented in Appendix D.1.

Table 1: **Summary of Rollout Relative** $L^2$ **Error (scaled by std)** across different methods and datasets. The best results are highlighted in bold. For flexible step rollout, step 3 works best for the PDEArena-Incomp dataset.

| Rollout Relative $L^2$ Error | Case | Ours (single step) | Ours (flexible step) | DPOT | MPP |
|---|---|---|---|---|---|
| Last step [1e-2] | PDEArena-Incomp | 76.77 | 68.03 | **65.27** | 93.52 |
| | PDEBench-Comp-LowVis | **39.11** | **39.11** | 48.92 | 65.32 |
| | PDEBench-Comp-HighVis | **61.41** | **61.41** | 2866 | 185.3 |
| All average [1e-2] | PDEArena-Incomp | 56.27 | 48.50 | **41.20** | 55.95 |
| | PDEBench-Comp-LowVis | **27.08** | **27.08** | 37.72 | 46.68 |
| | PDEBench-Comp-HighVis | **30.06** | **30.06** | 821.9 | 72.37 |

## 5.1 Superior Performance on Long-term Rollout Predictions

As demonstrated in Figure 2(a), Table 1, and Figure 3, VICON consistently outperforms baseline methods on long-horizon predictions across all benchmarks—with the exception of DPOT on the PDEArena-Incomp dataset, where our performance is comparable. Overall, VICON achieves an average reduction in relative $L^2$ RMSE at the final timestep of 37.9% compared to DPOT and 44.7% compared to MPP.

Notably, DPOT exhibits exceptionally poor performance on the PDEBench-Comp-HighVis dataset, with an 821.9% error compared to our 30.06%. Our visualization of failure cases reveals that DPOT struggles with trajectories with small pressure values compared to the dataset average. This may stems from DPOT's lack of prompt normalization, as compressible flow's pressure channel exhibit large magnitude variations.

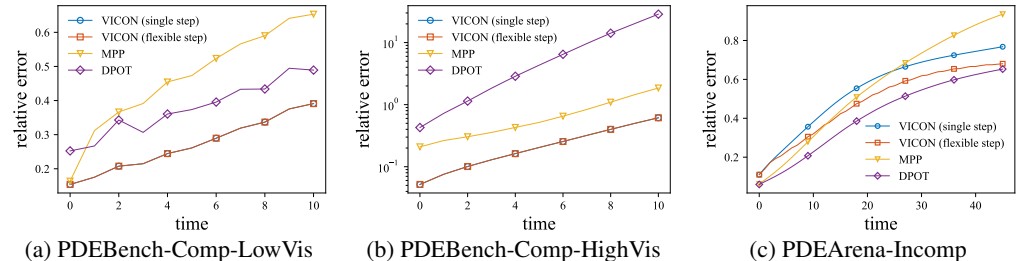

(a) PDEBench-Comp-LowVis     (b) PDEBench-Comp-HighVis     (c) PDEArena-Incomp

Figure 3: **Comparison of rollout errors (scaled by std) across different datasets and models.** We show errors for two VICON rollout strategies: single step rollout and flexible rollout strategy. For flexible step, step 3 works optimally for the PDEArena-Incomp dataset, while step 1 works best for PDEBench-Comp-LowVis and PDEBench-Comp-HighVis.

For the PDEArena-Incomp dataset, while VICON slightly underperforms DPOT and MPP initially, it quickly surpasses MPP and achieves comparable performance to DPOT for longer-step rollouts. This demonstrates VICON's robustness in long-term predictions. Despite DPOT's marginally better performance here, its poor performance on the PDEBench-Comp-HighVis dataset limits its applicability in multi-physics settings.

More detailed results are provided in Table 8 and Table 9 in Appendix D.

## 5.2 Flexible Rollout Strategies

As demonstrated in Figure 2(b) and Figure 5, VICON can select appropriate timestep strides based on each dataset's characteristics. While PDEBench-Comp-LowVis and PDEBench-Comp-HighVis perform best with single-step rollout, PDEArena-Incomp achieves optimal results with a stride of 3 using our flexible-step rollout strategy.

This dataset-dependent performance is expected: PDEBench-Comp-LowVis and PDEBench-Comp-HighVis record data with a larger timestep size and fewer total frames, making multi-stride predictions challenging. Conversely, PDEArena-Incomp records data with smaller timesteps and benefits from multi-stride prediction, as this approach reduces the rollout steps and error accumulation. As a result, VICON achieves an 11.4% error reduction in PDEArena-Incomp by selecting a balanced stride ($s_{\max} = 3$).

When evaluated with unseen strides ($s_{\max} = 7$), VICON maintains comparable performance to single-step strategies for PDEArena-Incomp, indicating that it extracts underlying operators through context pairs rather than memorizing dynamics. This generalization capability provides tolerance when deployed to real-world settings where device sampling rates often differ from a fixed training set—an advantage we further demonstrate in the next section.

## 5.3 Robustness to Imperfect Temporal Measurements

Real-world experimental measurements frequently suffer from imperfections—sampling rates may differ from training set, and frames can be missing due to device errors. Sequence-to-sequence models like MPP and DPOT require either retraining with new data or interpolation that introduces noise. In contrast, VICON elegantly handles such imperfections by forming context pairs from available frames only (Figure 1(c)). The algorithm for generating pairs with imperfect measurements appears in Appendix C.2.

We evaluate VICON against baselines on all benchmarks under two scenarios: (1) half sampling rate (dropping every other frame) and (2) random frame dropping (removing 1-3 frames per sequence). Results for the PDEBench-Comp-LowVis dataset in Figure 2(c) (with complete results in Figure 8) show that VICON experiences only 24.41% relative performance degradation compared to 71.37%-74.49% in baselines—demonstrating remarkable robustness to measurement imperfections.

## 5.4 Turbulence Kinetic Energy Analysis

Turbulence kinetic energy (TKE), defined as $\frac{1}{2}\left(\overline{\tilde{u}_x^2} + \overline{\tilde{u}_y^2}\right)$, is a critical metric for quantifying a model's ability to capture turbulent flow characteristics. Here, $\tilde{\boldsymbol{u}} = \boldsymbol{u} - \overline{\boldsymbol{u}}$ denotes the fluctuation of velocity from its statistical equilibrium state $\overline{\boldsymbol{u}}$.

Within our datasets, only a subset of PDEBench-Comp-LowVis, specifically those initialized with fully developed turbulent fields (see Appendix D.5 of [49]), is suitable for TKE analysis. After filtering these entries, we compare the mean absolute error (MAE) of the TKE between VICON, DPOT, and MPP. Our approach achieves a TKE error of 0.016, significantly outperforming MPP's error of 0.049, while achieving performance comparable to DPOT's error of 0.012. Visualizations of TKE errors are presented in Figure 6 in Appendix D.

## 5.5 Benefits of Multi-Physics Joint Training

We empirically examine whether VICON benefits from a multi-physics training by comparing two strategies: (1) training a single model *jointly* on all three datasets versus (2) training *separate* models specialized for individual datasets. For fair comparison, we maintained identical batch sizes across all trainings while adjusting the steps for separate training to be slightly more than one-third of the joint training duration, ensuring **comparable total computational costs** between approaches.

As shown in Figure 7, joint training significantly outperforms separate training across all three datasets. We attribute this performance gain to the underlying physical similarities across these fluid dynamics problems, where exposure to diverse yet related flow patterns enhances the model's generalization ability.

## 5.6 Computational Efficiency

In Figure 2(d) and Table 11, we compare the computational resources required by VICON and baselines. Our model demonstrates superior efficiency across multiple metrics. Compared to MPP, VICON requires approximately one-third of the inference time per frame while using 75% of the total parameters. VICON also outperforms DPOT, requiring 28% fewer parameters and 28% less inference time.

## 6 Conclusion and Future Works

We present VICON, a vision-transformer-based in-context operator network that efficiently processes dense physical fields through patch-wise operations. VICON overcomes the computational burden of original in-context operator networks in higher spatial dimensions while preserving the flexibility to extract multiphysics dynamics from few-shot contexts.

Our comprehensive experiments demonstrate that VICON achieves superior performance in long-term predictions, reducing the averaged last-step rollout error by 37.9% compared to DPOT and 44.7% compared to MPP, while requiring only 72.5% and 34.8% of their respective inference times. The model supports flexible rollout strategies with varying timestep strides, enabling natural application to imperfect real-world measurements where sampling frequencies differ or frames are randomly dropped—scenarios where VICON experiences only 24.41% relative performance degradation compared to 71.37%-74.49% degradation in baseline models.

Despite these advances, several challenges remain for future investigation. We emperically showed in Section 5.5 the benefits of multi-physics training, yet scaling to larger and more diverse datasets presents practical challenges. Specifically, generating high-fidelity physics simulation data across multiple domains requires both substantial computational resources and specialized domain expertise. Furthermore, the current approach does not yet extend to 3D applications, as token sequence length would grow cubically, exceeding our computational budget. The channel-union approach is also limited when incorporating domains beyond fluid dynamics with fundamentally different state variables. Finally, adapting VICON to handle irregular domains such as graphs or meshes would broaden its applications to areas like solid mechanics and molecular dynamics. Addressing these challenges will advance the promising paradigm of in-context learning for a broader range of physical systems.

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

## A  Dataset Details

### A.1  PDEArena-Incomp Dataset

The incompressible Navier-Stokes dataset comes from PDEArena [1]. The data are generated from the equation

$$\partial_t \boldsymbol{u} + \boldsymbol{u} \cdot \nabla \boldsymbol{u} = -\nabla p + \mu \Delta \boldsymbol{u} + \boldsymbol{f}, \tag{9}$$
$$\nabla \cdot \boldsymbol{u} = 0. \tag{10}$$

The space-time domain is $[0, 32]^2 \times [18, 105]$ where $dt = 1.5$ and the space resolution is $128 \times 128$. A scalar particle field is being transported with the fluids. The velocity fields satisfy Dirichlet boundary condition, and the scalar field satisfies Neumann boundary condition. The forcing term $\boldsymbol{f}$ is randomly sampled. The quantities of interest are the velocities and the scalar particle field.

### A.2  PDEBench-Comp-HighVis and PDEBench-Comp-LowVis Datasets

The PDEBench-Comp-HighVis and PDEBench-Comp-LowVis datasets come from PDEBench Compressible Navier-Stokes dataset [49]. The data are generated from the equation

$$\partial_t \rho + \nabla \cdot (\rho \boldsymbol{u}) = 0, \tag{11}$$
$$\rho(\partial_t \boldsymbol{u} + \boldsymbol{u} \cdot \nabla \boldsymbol{u}) = -\nabla p + \eta \Delta \boldsymbol{u} + (\zeta + \eta/3)\nabla(\nabla \cdot \boldsymbol{u}), \tag{12}$$
$$\partial_t \left( \varepsilon + \frac{\rho u^2}{2} \right) = -\nabla \cdot \left( \left( \varepsilon + p + \frac{\rho u^2}{2} \right) \boldsymbol{u} - \boldsymbol{u} \cdot \sigma' \right). \tag{13}$$

The space-time domain is $\mathbb{T}^2 \times [0, 1]$ where $dt = 0.05$. The datasets contain different combinations of shear and bulk viscosities. We group the ones with larger viscosities into the PDEBench-Comp-HighVis dataset, and the ones with extremely small (1e-8) viscosities into the PDEBench-Comp-LowVis dataset. The PDEBench-Comp-HighVis dataset has space resolution $128 \times 128$. The PDEBench-Comp-LowVis dataset has raw space resolution $512 \times 512$ and is downsampled to $128 \times 128$ through average pooling for consistency. The quantities of interest are velocities, pressure, and density.

### A.3  QOI Union and Channel Mask

Since each dataset contains different sets of quantities of interest, we take their union to create a unified representation. The unified physical field has 7 channels in total, with the following ordering:

1. Density ($\rho$)

2. Velocity at x direction ($u_x$)

3. Velocity at y direction ($u_y$)

4. Pressure ($P$)

5. Vorticity ($\omega$)

6. Passively transported scalar field ($S$)

7. Node type indicator (0: interior node, 1: boundary node)

For each dataset, we use a channel mask to indicate its valid fields and only calculate loss on these channels. The node type channel is universally excluded from loss calculations across all datasets.

## B  Experiment Details

### B.1  VICON Model Details

Here we provide key architectural parameters of VICON model implementation in Table 2.

Table 2: Model Configuration Details

| | |
|---|---|
| *Patch Configuration* | |
| Input patch numbers | $8 \times 8$ |
| Output patch numbers | $8 \times 8$ |
| Patch resolution | $16 \times 16$ |
| *Positional Encodings Shapes* | |
| Patch positional encodings | $[64, 1024]$ |
| Function positional encodings | $[20, 1024]$ |
| *Transformer Configuration* | |
| Hidden dimension | 1024 |
| Number of attention heads | 8 |
| Feedforward dimension | 2048 |
| Number of layers | 10 |
| Dropout rate | 0.0 |
| Number of COND & QOI pairs | 10 |
| Number of QOI exempted from loss calculation | 5 |

Table 3: Optimization Hyperparameters

| Parameter | Value |
|---|---|
| *Learning Rate Schedule* | |
| Scheduler | Cosine Annealing with Linear Warmup |
| Peak learning rate | $1 \times 10^{-4}$ |
| Final learning rate | $1 \times 10^{-7}$ |
| Warmup steps | 20,000 |
| Total steps | 200,000 |
| *Optimization Settings* | |
| Optimizer | AdamW |
| Weight decay | $1 \times 10^{-4}$ |
| Gradient norm clip | 1.0 |

## B.2 Training Details

We implement our method in PyTorch [50] and utilize data parallel training [51] across two NVIDIA RTX 4090 GPUs. We employ the AdamW optimizer with a cosine learning rate schedule that includes a linear warmup phase. We apply gradient clipping with a maximum norm of 1.0 to ensure training stability. All optimization parameters are detailed in Table 3.

## B.3 MPP Details

For a fair comparison, we retrained MPP [26] using our training dataset using the same model configurations and optimizer hyperparameters (batch size is set to be the maximum possible on our device). We evaluate MPP using the same testing setup and metric.

## B.4 DPOT Details

For a fair comparison, we retrain DPOT [27] using our training dataset using the same model configurations and optimizer hyperparameters (batch size is set to be the maximum possible on our device). We evaluate DPOT using the same testing setup and metric.

---

**Algorithm 1:** GenerateSingleStepStrategy

---

**Input:** D: Number of demonstrations to use
R: Number of ground truth reference steps
T: Total steps in sequence
**Output:** S: Strategy list of example pairs and question pairs

```
1 S ← [ ]
2 Ei ← range(0, D)                    /* example input indices:  0,1,...,D-1 */
3 Eo ← range(1, D+1)                  /* example output indices: 1,2,...,D */
4 for i from R to T-1 do
5   │  Qi ← i - 1                     /* question input is previous frame */
6   │  Qo ← i                         /* question output is current frame */
7   │  S.append((Ei, Eo), Qi, Qo)
8 end
9 return S
```

---

## C  Algorithms and Examples of Strategy Generation

### C.1  Strategy with Full Temporal Sequence

**Algorithm for single-step strategy.**    Algorithm 1 presents our approach for generating single-step rollout strategies when all temporal frames are available. This strategy maintains a fixed stride of 1 between consecutive frames, providing stable but potentially error-accumulating predictions.

**Algorithm for flexible-step strategy.**    Algorithm 2 describes our approach for generating flexible-step strategies with variable strides up to a maximum value. This approach reduces the total number of rollout steps required, potentially mitigating error accumulation for long sequences.

**Rollout Strategy Example.**    We demonstrate two rollout strategies in Tables 4 and 5, showing single-step and flexible-step strategies, respectively. In both cases, the initial frames span from time step 0 to 9, and we aim to predict the trajectory up to time step 20.

As shown in Table 5, the flexible-step strategy initially uses smaller strides to build sufficient examples, then employs maximum strides for later rollouts, as detailed in Section 4.5. We note that repeated in-context examples appear in Table 5, which is common when the maximum stride is large and the initial frames cannot form enough examples. While the number of in-context examples in VICON is flexible and our model can accommodate fewer examples than the designed length, our preliminary experiments indicate that the model performs better with more in-context examples, even when some examples are repeated.

### C.2  Strategy with Imperfect Temporal Sequence

When facing imperfect temporal sampling where certain frames are missing, we can still form valid demonstration pairs from the available frames. Algorithm 3 presents this adaptive pair selection process, which serves as a fundamental building block for all strategy generation algorithms in scenarios with irregular temporal data.

**Algorithm for single-step strategy with drops.**    Algorithm 4 extends the single-step strategy to handle scenarios where frames are missing from the input sequence, adaptively forming strategies from available frames while maintaining the fixed stride constraint.

**Algorithm for flexible-step strategy.**    Algorithm 5 presents our solution for generating flexible-step rollout strategies when frames are missing, dynamically selecting appropriate strides based on available frames and previous predictions.

**Rollout Strategy Example with Missing Frames.**    Tables 6 and 7 illustrate our adaptive strategies when frames 2, 5, and 9 are missing from the initial sequence (with all other settings identical to those in Appendix C.1). The tables demonstrate single-step and flexible-step (max stride: 3)

**Algorithm 2:** GenerateFlexibleStepStrategy

---

**Input:** D: Number of demonstrations to use
R: Number of ground truth reference steps
M: Maximum stride between pairs
T: Total steps in sequence
**Output:** S: Strategy list of example pairs and question pairs
```
/* Require R >= M + 1 to ensure examples exist for each stride      */
```
1   $S \leftarrow [\ ]$
2   $Ec \leftarrow \{\}$               `/* example condition indices by stride */`
3   $Eq \leftarrow \{\}$               `/* example query indices by stride */`
   `/* Prepare example pairs for each stride                        */`
4   **for** *s from 1 to M* **do**
5      $Nd \leftarrow min(D, R - s)$        `/* available examples for this stride */`
6      $Eqo \leftarrow range(R - Nd, R)$        `/* output indices for examples */`
       `/* If we need more examples than available, repeat them          */`
7      $reps \leftarrow \lceil D / Nd \rceil$        `/* ceiling division */`
8      $Eqo \leftarrow repeat(Eqo, reps)[0{:}D]$      `/* repeat and truncate to D */`
9      $Eqo.sort()$        `/* ensure increasing order */`
10     $Ec[s] \leftarrow Eqo - s$        `/* input indices are s steps before outputs */`
11     $Eq[s] \leftarrow Eqo$        `/* store output indices */`
12   **end**
   `/* Generate rollout strategy                                     */`
13   **for** *i from R to T-1* **do**
14     $dist \leftarrow i - R + 1$        `/* distance from last reference frame */`
15     $s \leftarrow min(dist, M)$        `/* select appropriate stride */`
16     $Qi \leftarrow i - s$        `/* question input index */`
17     $Qo \leftarrow i$        `/* question output index */`
18     $S.append((Ec[s], Eq[s]), Qi, Qo)$
19   **end**
20   **return** S

---

Table 4: Single-step Rollout Strategy Example

| Rollout index | Examples (COND, QOI) | Question COND | Predict QOI |
|---|---|---|---|
| 1 | (0,1) (1,2) (2,3) (3,4) (4,5) (5,6) (6,7) (7,8) (8,9) | 9 | 10 |
| 2 | (0,1) (1,2) (2,3) (3,4) (4,5) (5,6) (6,7) (7,8) (8,9) | 10 | 11 |
| 3 | (0,1) (1,2) (2,3) (3,4) (4,5) (5,6) (6,7) (7,8) (8,9) | 11 | 12 |
| 4 | (0,1) (1,2) (2,3) (3,4) (4,5) (5,6) (6,7) (7,8) (8,9) | 12 | 13 |
| 5 | (0,1) (1,2) (2,3) (3,4) (4,5) (5,6) (6,7) (7,8) (8,9) | 13 | 14 |
| 6 | (0,1) (1,2) (2,3) (3,4) (4,5) (5,6) (6,7) (7,8) (8,9) | 14 | 15 |
| 7 | (0,1) (1,2) (2,3) (3,4) (4,5) (5,6) (6,7) (7,8) (8,9) | 15 | 16 |
| 8 | (0,1) (1,2) (2,3) (3,4) (4,5) (5,6) (6,7) (7,8) (8,9) | 16 | 17 |
| 9 | (0,1) (1,2) (2,3) (3,4) (4,5) (5,6) (6,7) (7,8) (8,9) | 17 | 18 |
| 10 | (0,1) (1,2) (2,3) (3,4) (4,5) (5,6) (6,7) (7,8) (8,9) | 18 | 19 |
| 11 | (0,1) (1,2) (2,3) (3,4) (4,5) (5,6) (6,7) (7,8) (8,9) | 19 | 20 |

approaches, respectively, highlighting how our algorithm dynamically selects appropriate example pairs to maintain prediction capability despite missing some temporal samplings.

# D   Additional Experimental Results.

## D.1   Ablation Studies

**Impact of Patch Resolutions.**   We conducted ablation studies on patch resolution by varying patch sizes (8, 16, 32, 64) to find a balance between spatial granularity and computational resource constraint. While smaller patches theoretically capture finer details, they generate longer token sequences, hitting memory caps due to transformer's quadratic complexity. For patches smaller

Table 5: Flexible-step Rollout Strategy Example ($s_{\max} = 5$)

| Rollout index | Examples (COND, QOI) | Question COND | Predict QOI |
|---|---|---|---|
| 1 | (0,1) (1,2) (2,3) (3,4) (4,5) (5,6) (6,7) (7,8) (8,9) | 9 | 10 |
| 2 | (0,2) (0,2) (1,3) (2,4) (3,5) (4,6) (5,7) (6,8) (7,9) | 9 | 11 |
| 3 | (0,3) (0,3) (1,4) (1,4) (2,5) (3,6) (4,7) (5,8) (6,9) | 9 | 12 |
| 4 | (0,4) (0,4) (1,5) (1,5) (2,6) (2,6) (3,7) (4,8) (5,9) | 9 | 13 |
| 5 | (0,5) (0,5) (1,6) (1,6) (2,7) (2,7) (3,8) (3,8) (4,9) | 9 | 14 |
| 6 | (0,5) (0,5) (1,6) (1,6) (2,7) (2,7) (3,8) (3,8) (4,9) | 10 | 15 |
| 7 | (0,5) (0,5) (1,6) (1,6) (2,7) (2,7) (3,8) (3,8) (4,9) | 11 | 16 |
| 8 | (0,5) (0,5) (1,6) (1,6) (2,7) (2,7) (3,8) (3,8) (4,9) | 12 | 17 |
| 9 | (0,5) (0,5) (1,6) (1,6) (2,7) (2,7) (3,8) (3,8) (4,9) | 13 | 18 |
| 10 | (0,5) (0,5) (1,6) (1,6) (2,7) (2,7) (3,8) (3,8) (4,9) | 14 | 19 |
| 11 | (0,5) (0,5) (1,6) (1,6) (2,7) (2,7) (3,8) (3,8) (4,9) | 15 | 20 |

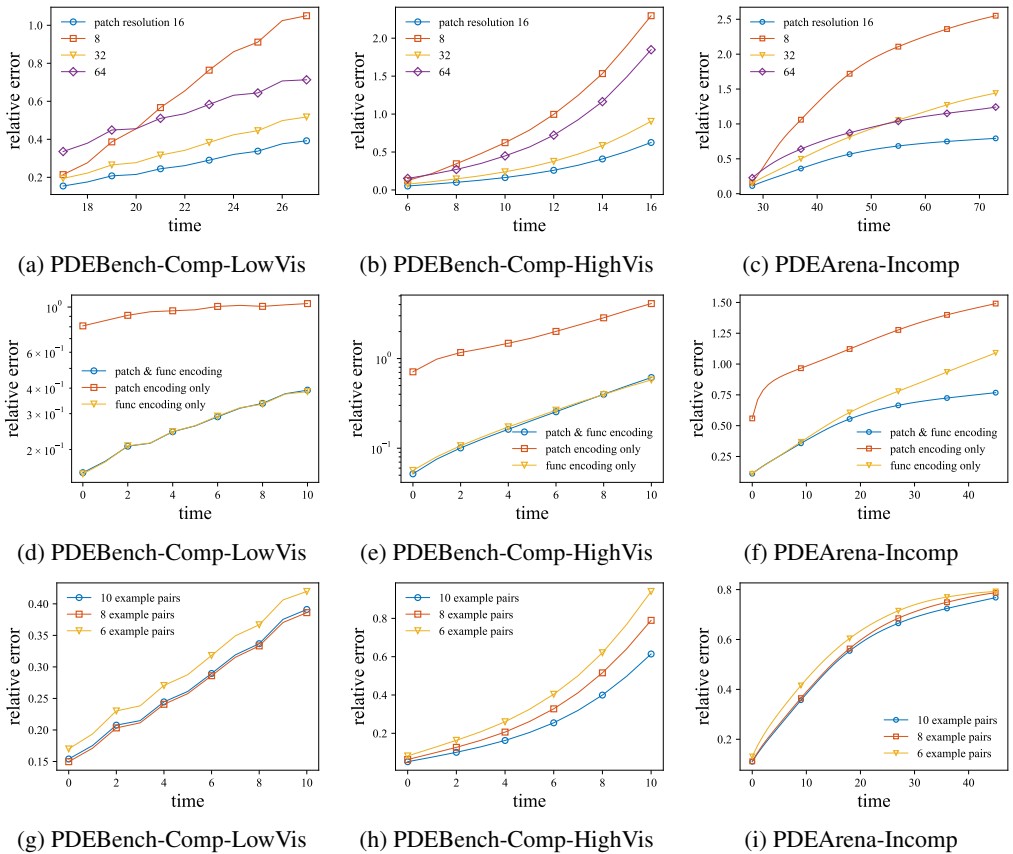

(a) PDEBench-Comp-LowVis    (b) PDEBench-Comp-HighVis    (c) PDEArena-Incomp

(d) PDEBench-Comp-LowVis    (e) PDEBench-Comp-HighVis    (f) PDEArena-Incomp

(g) PDEBench-Comp-LowVis    (h) PDEBench-Comp-HighVis    (i) PDEArena-Incomp

Figure 4: Ablation studies across the three datasets. **Top row (a-c):** Impact of patch resolutions $(8, 16, 32, 64)$ showing optimal performance at patch size 16. **Middle row (d-f):** Effect of different positional encoding combinations. **Bottom row (g-i):** Performance variation with different context lengths (6, 8, 10 pairs).

**Algorithm 3:** GetAvailablePairs

**Input:** D: Number of demonstrations needed
dt: Desired time stride between pairs
Fa: List of indices of available frames
**Output:** P: List of input-output index pairs with the specified stride

```
1 Fa.sort()
2 P ← [ ]                                    /* Initialize empty pairs list */
  /* Find all pairs with stride dt                                        */
3 for i from 0 to len(Fa) - 1 do
4     for j from i + 1 to len(Fa) - 1 do
5         if Fa[j] - Fa[i] == dt then
6             │ P.append((Fa[i], Fa[j]))
7         end
8     end
9 end
10 if P is empty then
11     return [ ]
12 end
   /* If we don't have enough unique pairs, repeat them                   */
13 if len(P) < D then
14     reps ← ⌈D / len(P)⌉                            /* ceiling division */
15     RP ← repeat(P, reps)[0:D]            /* repeat pairs and truncate to D */
16     P ← RP
17 end
18 return P
```

than $16 \times 16$, we had to reduce token dimensions ($1024{\rightarrow}512$) and feedforward dimensions (from 2048 to 1024), while patches below $8 \times 8$ remained infeasible even with mixed precision training. Figure 4(a-c) shows that patch size 16 achieves optimal performance across all datasets—balancing sequence length and representational capacity. Performance degradation with coarser patches (32, 64) aligns with expectations, while degradation with finer patches stems from the necessary reduction in hidden dimensions, highlighting a fundamental challenge in scaling to 3D applications.

**Positional Encodings.** We evaluated different combinations of positional encodings as shown in Figure 4(d-f). Our architecture employs two types of encodings: patch position encodings (for spatial relationships between patches) and function encodings (differentiating input/output in different pairs) (Section 4.2).

The results demonstrate that including both encoding types consistently yields the best performance across all datasets. Notably, function encodings have a more significant impact than patch encodings, highlighting the importance of distinguishing between the condition and qoi beyond what causal masking (equation 6) provides.

The impact of patch encodings varies across datasets: they show minimal effect on the PDEBench datasets (PDEBench-Comp-LowVisand PDEBench-Comp-HighVis) yet significantly improve performance on PDEArena-Incomp. We attribute this to the different stiffness of these systems. In compressible flows (PDEBench), spatial interactions naturally decay with distance, creating consistent and monotonic spatial correlations between patches. In contrast, Navier-Stokes equations exhibit infinite stiffness where correlations between any spatial locations are instantaneous and determined by the global velocity field, creating non-monotonic and case-dependant relationships. Explicitly encoding spatial positions therefore becomes particularly beneficial for PDEArena-Incomp, as it reduces the learning complexity for these non-local dependencies.

**Varying Context Length.** We investigated how the number of in-context examples affects model performance, as shown in Figure 4(g-i). Following insights from the original ICON work [25], we evaluated context lengths of 6, 8, and 10 pairs, balancing performance against computational efficiency. The results show that 10 in-context examples deliver significantly better results for

**Algorithm 4:** GenerateSingleStepStrategyWithDrops

**Input:** D: Number of demonstrations to use
S: Fixed stride between pairs
T: Total steps in sequence
Fa: List of indices of available frames
**Output:** St: Strategy list of (E, Qi, Qo) tuples

```
 1 St ← [ ]
 2 Fa.sort()
 3 Fs ← Fa[-1]                                    /* the starting frame for rollout */
 4 P ← GetAvailablePairs(D, S, Fa)                    /* pairs with fixed stride */
 5 if P is empty then
 6 │   return [ ]              /* No available pairs with the specified stride */
 7 end
 8 Fc ← Fa.copy()                  /* accumulated frames (current + predicted) */
 9 for i from (Fs + 1) to (T - 1) do
10 │   Ci ← i - S                              /* potential condition index */
11 │   if Ci ∉ Fc then
12 │   │   continue                    /* Cannot predict frame i with stride S */
13 │   end
14 │   Pdt ← P[s]
   │                              /* The example pairs for predicting frame i */
15 │   Qi ← i - s                                  /* question input index */
16 │   Qo ← i                                      /* question output index */
17 │   S.append(Pdt, Qi, Qo)
18 │   Fc.append(Qo)      /* The predicted frame can be used for future steps */
19 end
20 return St
```

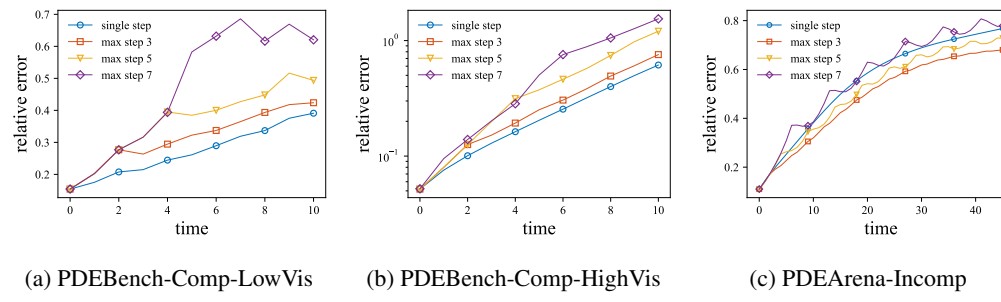

(a) PDEBench-Comp-LowVis    (b) PDEBench-Comp-HighVis    (c) PDEArena-Incomp

Figure 5: Comparison of rollout errors across different datasets, using single-step and flexible-step strategies with varying maximum step sizes ($s_{\max} = 1, 3, 5, 7$).

PDEBench-Comp-LowVis compared to shorter contexts, while for the remaining datasets, 10 and 8 in-context examples perform similarly (both outperforming 6 examples).

## D.2 More Results and Visualizations

**Detailed Results.** Tables 8 and 9 summarize the relative and absolute $L^2$ rollout errors across different timesteps for all evaluated models on the three datasets.

**Generalization to different timestep strides.** Figure 5 illustrates VICON's performance with varying timestep strides ($s_{\max} = 1, 3, 5, 7$) across all three datasets, demonstrating the model's ability to adapt to different temporal resolutions without retraining.

**Turbulence Kinetic Energy (TKE) Predictions.** Figure 6 displays the visualization of TKE fields for ground truth and model predictions, highlighting VICON's superior ability to preserve critical turbulent flow structures compared to baseline approaches.

**Algorithm 5:** GenerateFlexibleStepStrategyWithDrops

---

**Input:** D: Number of demonstrations to use
M: Maximum stride between pairs for rollout
T: Total steps in the sequence
Fa: List of indices of available frames
**Output:** S: Strategy list of (E, Qi, Qo) tuples, where E are the list of example pair indices (Ei,
        Eo) and Qi, Qo are the question and output indices

---

```
1  S ← [ ]
2  Fa.sort()
3  Fs ← Fa[-1]                                    /* the starting frame for rollout */
4  P ← {}                                          /* dictionary of available pairs */
5  for dt from 1 to M do
6  │   Pdt ← GetAvailablePairs(D, dt, Fa)          /* pairs with stride dt in Fa */
7  │   if Pdt is not empty then
8  │   │   P[dt] ← Pdt
9  │   end
10 end
11 Ms ← max(P.keys())                              /* maximum available stride */
12 Fc ← Fa.copy()                    /* accumulated frames (current + predicted) */
13 for i from (Fs + 1) to (T - 1) do
14 │   dt ← i - Fs
15 │   Mt ← min(dt, M, Ms)                         /* maximum stride for current step */
16 │   found ← False
17 │   for s in P.keys().sorted(reverse=True) where s ≤ Mt do
18 │   │   Ci ← i - s                              /* potential condition index */
19 │   │   if Ci ∈ Fc then
       │   │       /* Found a starting index to obtain frame i              */
20 │   │   │   found ← True
21 │   │   │   break
22 │   │   end
23 │   end
24 │   if not found then
       │       /* We cannot predict frame i                                 */
25 │   │   continue
26 │   end
       │   /* The example pairs for predicting frame i                      */
27 │   Pdt ← P[s]
28 │   Qi ← i - s                                  /* question input index */
29 │   Qo ← i                                      /* question output index */
30 │   S.append(Pdt, Qi, Qo)
31 │   Fc.append(Qo)
       │       /* The Qo frame is obtained and can be used for future rollouts */
32 end
33 return S
```

---

604 **Comparison between joint training vs separate training.** Figure 7 and Table 10 quantify the
605 performance differences between jointly trained and separately trained models, demonstrating the
606 significant advantages of multi-physics training across all datasets.

607 **Computational Efficiency.** Table 11 summarizes the computational resources required by each
608 method, showing VICON's advantages in terms of training cost, inference speed, and model parameter
609 count compared to both DPOT and MPP baselines.

610 **Rollout with imperfect measurements.** Figure 8 show the rollout results with imperfect temporal
611 measurements.

Table 6: Single-step Rollout Strategy Example with Missing Frames

| Rollout index | Examples (COND, QOI) | Question COND | Predict QOI |
|---|---|---|---|
| 1 | (0,1) (0,1) (0,1) (3,4) (3,4) (6,7) (6,7) (7,8) (7,8) | 8 | 9 |
| 2 | (0,1) (0,1) (0,1) (3,4) (3,4) (6,7) (6,7) (7,8) (7,8) | 9 | 10 |
| 3 | (0,1) (0,1) (0,1) (3,4) (3,4) (6,7) (6,7) (7,8) (7,8) | 10 | 11 |
| 4 | (0,1) (0,1) (0,1) (3,4) (3,4) (6,7) (6,7) (7,8) (7,8) | 11 | 12 |
| 5 | (0,1) (0,1) (0,1) (3,4) (3,4) (6,7) (6,7) (7,8) (7,8) | 12 | 13 |
| 6 | (0,1) (0,1) (0,1) (3,4) (3,4) (6,7) (6,7) (7,8) (7,8) | 13 | 14 |
| 7 | (0,1) (0,1) (0,1) (3,4) (3,4) (6,7) (6,7) (7,8) (7,8) | 14 | 15 |
| 8 | (0,1) (0,1) (0,1) (3,4) (3,4) (6,7) (6,7) (7,8) (7,8) | 15 | 16 |
| 9 | (0,1) (0,1) (0,1) (3,4) (3,4) (6,7) (6,7) (7,8) (7,8) | 16 | 17 |
| 10 | (0,1) (0,1) (0,1) (3,4) (3,4) (6,7) (6,7) (7,8) (7,8) | 17 | 18 |
| 11 | (0,1) (0,1) (0,1) (3,4) (3,4) (6,7) (6,7) (7,8) (7,8) | 18 | 19 |

Table 7: Flexible-step Rollout Strategy Example with Missing Frames (max stride: 3)

| Rollout index | Examples (COND, QOI) | Question COND | Predict QOI |
|---|---|---|---|
| 1 | (0,1) (0,1) (0,1) (3,4) (3,4) (6,7) (6,7) (7,8) (7,8) | 8 | 9 |
| 2 | (1,3) (1,3) (1,3) (4,6) (4,6) (4,6) (6,8) (6,8) (6,8) | 8 | 10 |
| 3 | (0,3) (0,3) (0,3) (1,4) (1,4) (3,6) (3,6) (4,7) (4,7) | 8 | 11 |
| 4 | (0,3) (0,3) (0,3) (1,4) (1,4) (3,6) (3,6) (4,7) (4,7) | 9 | 12 |
| 5 | (0,3) (0,3) (0,3) (1,4) (1,4) (3,6) (3,6) (4,7) (4,7) | 10 | 13 |
| 6 | (0,3) (0,3) (0,3) (1,4) (1,4) (3,6) (3,6) (4,7) (4,7) | 11 | 14 |
| 7 | (0,3) (0,3) (0,3) (1,4) (1,4) (3,6) (3,6) (4,7) (4,7) | 12 | 15 |
| 8 | (0,3) (0,3) (0,3) (1,4) (1,4) (3,6) (3,6) (4,7) (4,7) | 13 | 16 |
| 9 | (0,3) (0,3) (0,3) (1,4) (1,4) (3,6) (3,6) (4,7) (4,7) | 14 | 17 |
| 10 | (0,3) (0,3) (0,3) (1,4) (1,4) (3,6) (3,6) (4,7) (4,7) | 15 | 18 |
| 11 | (0,3) (0,3) (0,3) (1,4) (1,4) (3,6) (3,6) (4,7) (4,7) | 16 | 19 |

Table 8: **(Comparison) Summary of Rollout Relative $L^2$ Error (scale by std)** for different methods across various cases. The best results are highlighted in bold.

| Rollout Relative $L^2$ Error | Case | Ours (single step) | Ours (flexible step) | DPOT | MPP |
|---|---|---|---|---|---|
| Step 1 [1e-2] | PDEArena-Incomp | 11.01 | 11.01 | **5.97** | 6.17 |
| | PDEBench-Comp-LowVis | **15.40** | **15.40** | 25.24 | 16.37 |
| | PDEBench-Comp-HighVis | **5.18** | **5.18** | 42.76 | 20.90 |
| Step 5 [1e-2] | PDEArena-Incomp | 22.68 | 20.62 | **11.70** | 14.81 |
| | PDEBench-Comp-LowVis | **24.45** | **24.45** | 36.05 | 45.45 |
| | PDEBench-Comp-HighVis | **16.25** | **16.25** | 286.3 | 42.93 |
| Step 10 [1e-2] | PDEArena-Incomp | 35.67 | 30.53 | **20.74** | 27.89 |
| | PDEBench-Comp-LowVis | **37.54** | **37.54** | 49.47 | 64.11 |
| | PDEBench-Comp-HighVis | **49.83** | **49.83** | 2016 | 144.46 |
| Last step [1e-2] | PDEArena-Incomp | 76.77 | 68.03 | **65.27** | 93.52 |
| | PDEBench-Comp-LowVis | **39.11** | **39.11** | 48.92 | 65.32 |
| | PDEBench-Comp-HighVis | **61.41** | **61.41** | 2866 | 185.3 |
| All average [1e-2] | PDEArena-Incomp | 56.27 | 48.50 | **41.20** | 55.95 |
| | PDEBench-Comp-LowVis | **27.08** | **27.08** | 37.72 | 46.68 |
| | PDEBench-Comp-HighVis | **30.06** | **30.06** | 821.9 | 72.37 |

**Visualizations.** We compare the output of different models in Figure 9, Figure 10, and Figure 11. Figures 12 and 13 present additional visualizations of the VICON model outputs compared to ground truth and baseline predictions, highlighting the qualitative advantages of our approach.

Table 9: **(Comparison) Summary of Rollout Absolute $L^2$ Error** for different methods across various cases. The best results are highlighted in bold.

| Rollout $L^2$ Error | Case | Ours (single step) | Ours (flexible step) | DPOT | MPP |
|---|---|---|---|---|---|
| Step 1 [1e-2] | PDEArena-Incomp | 5.63 | 5.63 | **3.02** | 3.12 |
| | PDEBench-Comp-LowVis | **21.74** | **21.74** | 29.47 | 24.47 |
| | PDEBench-Comp-HighVis | **1.43** | **1.43** | 6.22 | 4.73 |
| Step 5 [1e-2] | PDEArena-Incomp | 10.38 | 9.46 | **5.37** | 6.79 |
| | PDEBench-Comp-LowVis | **31.23** | **31.23** | 43.57 | 57.50 |
| | PDEBench-Comp-HighVis | **2.34** | **2.34** | 14.48 | 6.52 |
| Step 10 [1e-2] | PDEArena-Incomp | 14.65 | 12.55 | **8.55** | 11.49 |
| | PDEBench-Comp-LowVis | **45.39** | **45.39** | 59.43 | 78.54 |
| | PDEBench-Comp-HighVis | **3.21** | **3.21** | 25.79 | 8.98 |
| Last Step [1e-2] | PDEArena-Incomp | 16.50 | 14.56 | **14.03** | 19.47 |
| | PDEBench-Comp-LowVis | **48.69** | **48.69** | 63.06 | 85.60 |
| | PDEBench-Comp-HighVis | **3.39** | **3.39** | 27.98 | 9.56 |
| All average [1e-2] | PDEArena-Incomp | 16.26 | 14.31 | **11.86** | 15.77 |
| | PDEBench-Comp-LowVis | **34.44** | **34.44** | 46.60 | 60.68 |
| | PDEBench-Comp-HighVis | **2.48** | **2.48** | 16.86 | 7.04 |

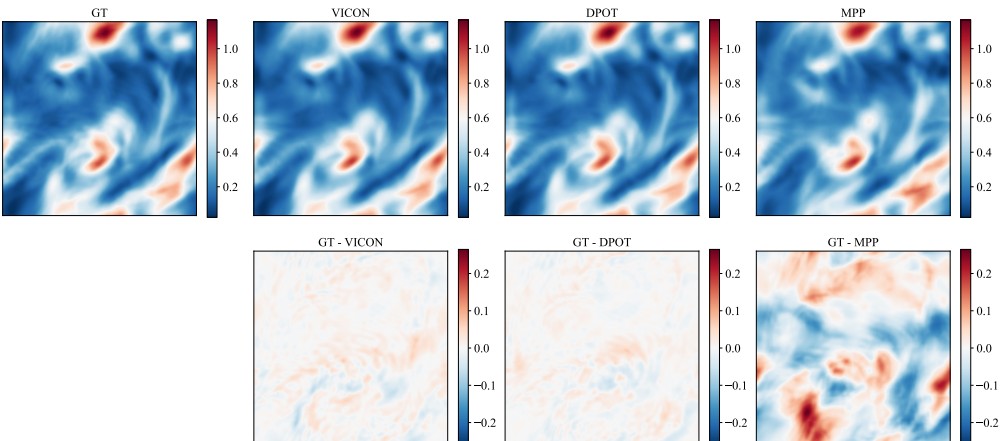

Figure 6: **Comparison of turbulence kinetic energy predictions.** (Top-left) Ground truth TKE field, (Top-right) model predictions, (Bottom) model error.

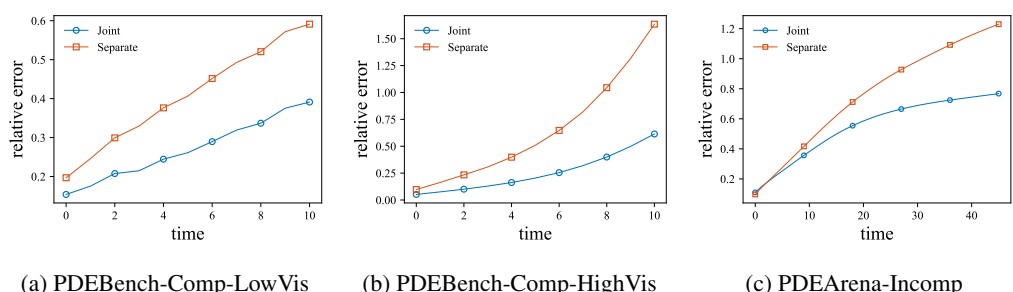

(a) PDEBench-Comp-LowVis    (b) PDEBench-Comp-HighVis    (c) PDEArena-Incomp

Figure 7: **Comparing rollout errors (single step, scale by std) for joint versus separate training strategies**. For separate training (individual models for each dataset), we maintain the same batch sizes as in joint training while adjusting training steps to be slightly more than one-third of the joint training duration, ensuring comparable total computational costs across both approaches.

Table 10: **Summary of Rollout Relative $L^2$ Error Metrics (single step, scale by std) for 2 Training Strategies**. The best results are highlighted in bold. For seperate training (a model on each dataset), each run's batch size is controlled to be the same as joint training. To ensure fair comparison, we maintain the same batch sizes as joint training for each seperate training runs while adjusting their training steps to be slightly more than one-third of the joint training duration, ensuring comparable total computational costs.

| Rollout Relative $L^2$ Error [1e-2] | Case | Joint | Seperate |
|---|---|---|---|
| | PDEArena-Incomp | 11.10 | **9.66** |
| Step 1 | PDEBench-Comp-LowVis | **15.61** | 19.68 |
| | PDEBench-Comp-HighVis | **5.79** | 9.74 |
| | PDEArena-Incomp | **23.00** | 24.09 |
| Step 5 | PDEBench-Comp-LowVis | **24.56** | 37.63 |
| | PDEBench-Comp-HighVis | **19.73** | 39.81 |
| | PDEArena-Incomp | **36.18** | 41.66 |
| Step 10 | PDEBench-Comp-LowVis | **37.47** | 57.17 |
| | PDEBench-Comp-HighVis | **57.88** | 131.6 |
| | PDEArena-Incomp | **77.81** | 123.0 |
| Last Step | PDEBench-Comp-LowVis | **39.03** | 59.11 |
| | PDEBench-Comp-HighVis | **71.17** | 163.5 |
| | PDEArena-Incomp | **56.26** | 76.35 |
| All average | PDEBench-Comp-LowVis | **27.08** | 40.75 |
| | PDEBench-Comp-HighVis | **30.06** | 65.19 |

Table 11: **(Comparison) Summary of Resource and Timing Metrics** for different methods.

| Resource and Timing Metrics | Ours | DPOT | MPP |
|---|---|---|---|
| Training cost [GPU hrs] | 58 | 70 | 64 |
| Rollout time per step [ms] | 8.7 | 12.0 | 25.7 |
| Model Param Size | 88M | 122M | 116M |

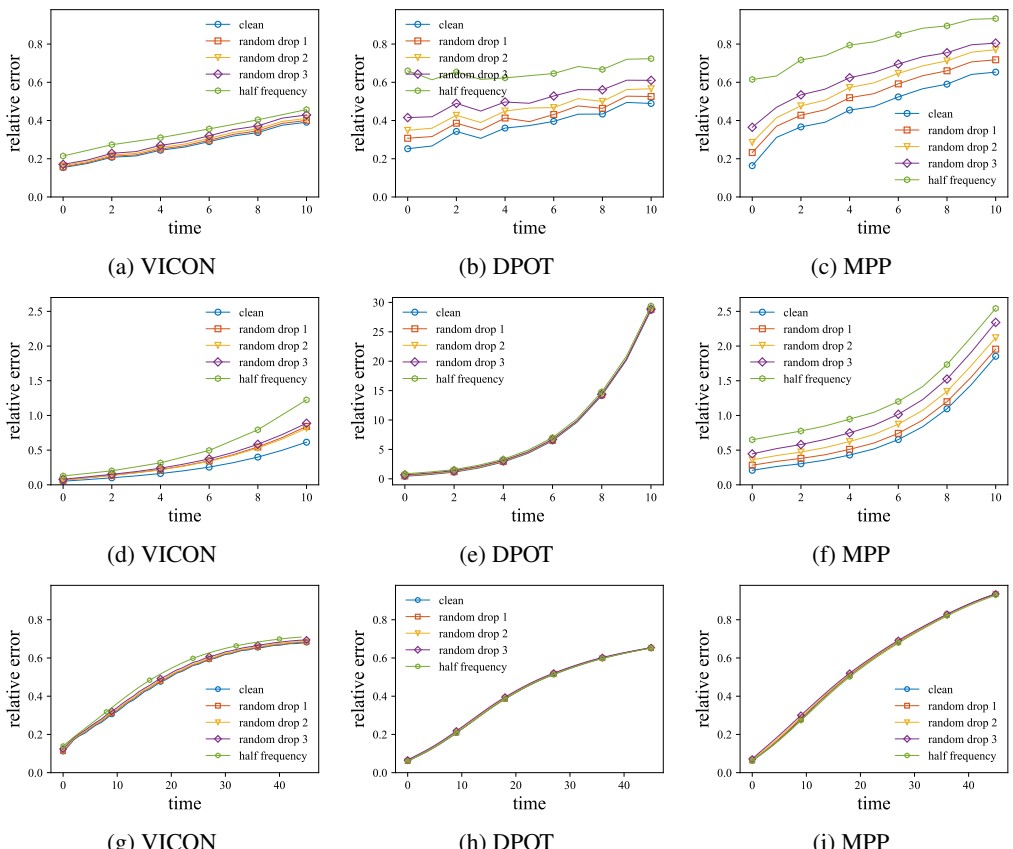

Figure 8: **Comparison of rollout errors (scale by std) with different input data noise levels across all datasets.** Each column (left to right) shows VICON, DPOT, and MPP models, while each row (top to bottom) represents PDEBench-Comp-LowVis, PDEBench-Comp-HighVis, and PDEArena-Incomp datasets. For MPP and DPOT which require fixed $dt$ and context window, interpolation is used to generate missing frames, while VICON can directly handle irregular temporal data without interpolation.

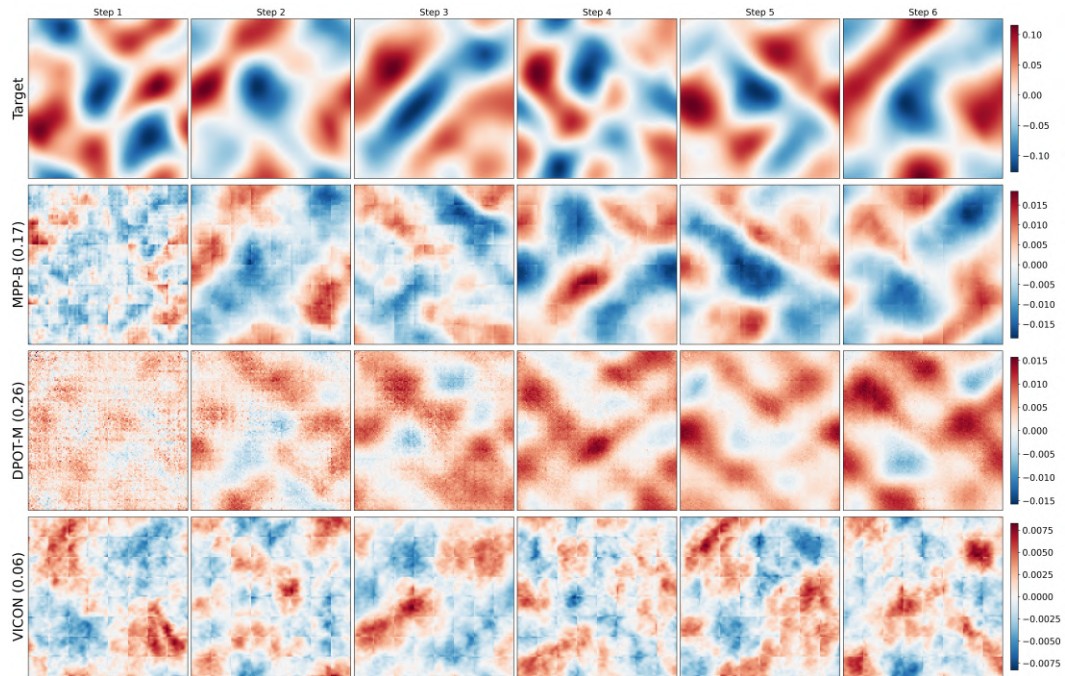

Figure 9: **Comparing outputs from different models.** The target is the first 6 output steps from PDEBench-Comp-HighVis dataset (x-velocity channel). For each model(each row), we display the difference between target and model output. Errors (rescale by std) for the full trajectory are listed after the model names.

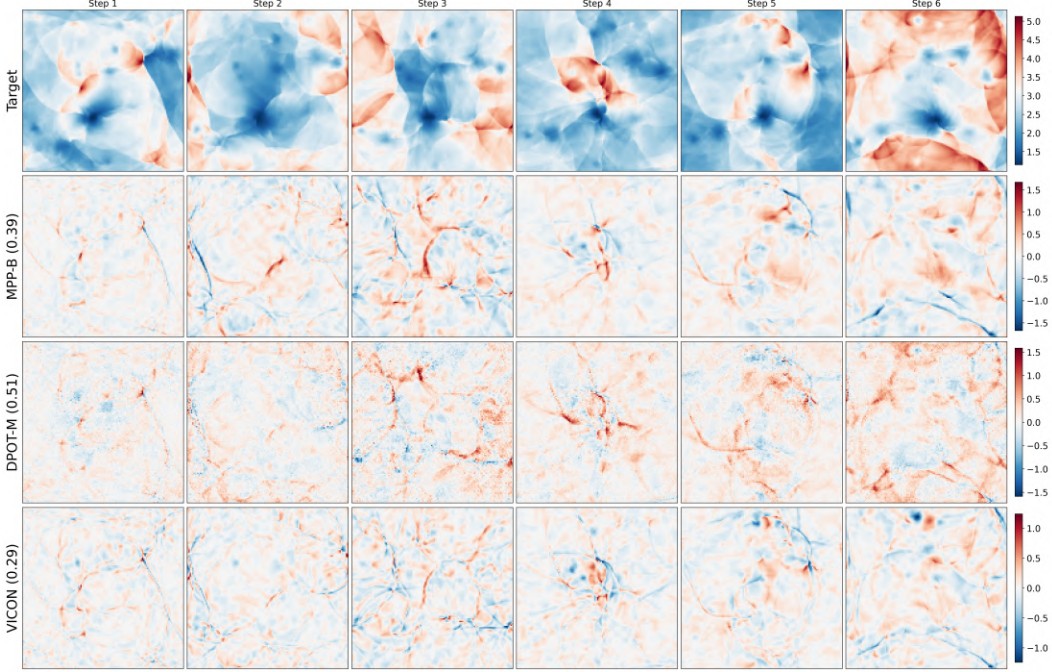

Figure 10: **Comparing outputs from different models.** The target is the first 6 output steps from PDEBench-Comp-LowVis dataset (pressure channel). For each model(each row), we display the difference between target and model output. Errors (rescale by std) for the full trajectory are listed after the model names.

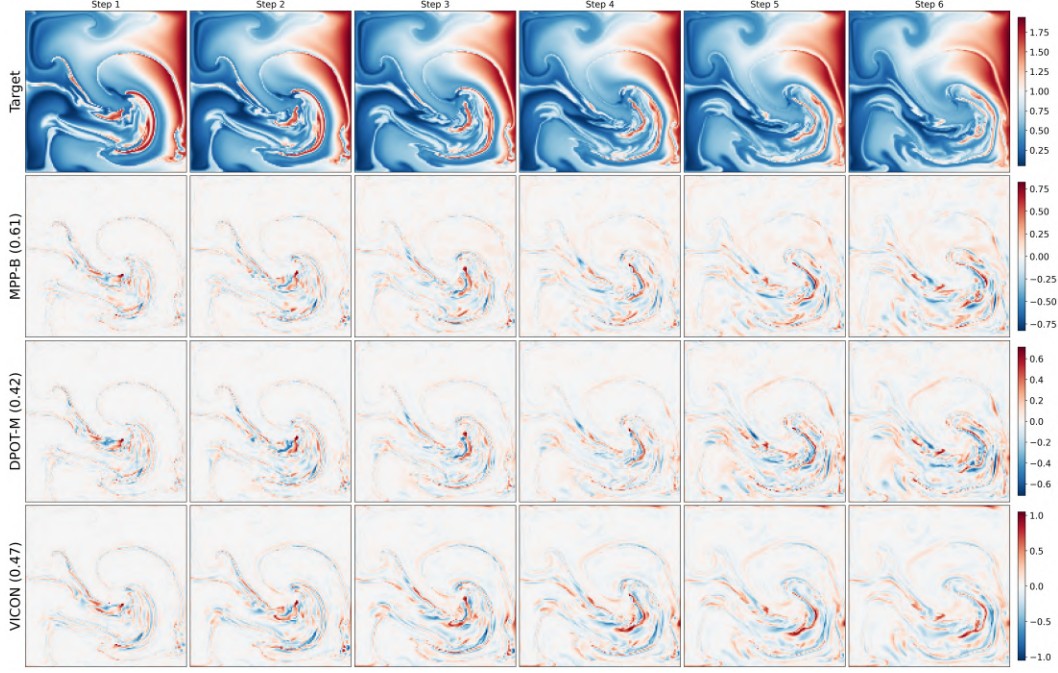

Figure 11: **Comparing outputs from different models.** The target is the first 6 output steps from PDEArena-Incomp dataset (particle density channel). For each model(each row), we display the difference between target and model output. Errors (rescale by std) for the full trajectory are listed after the model names.

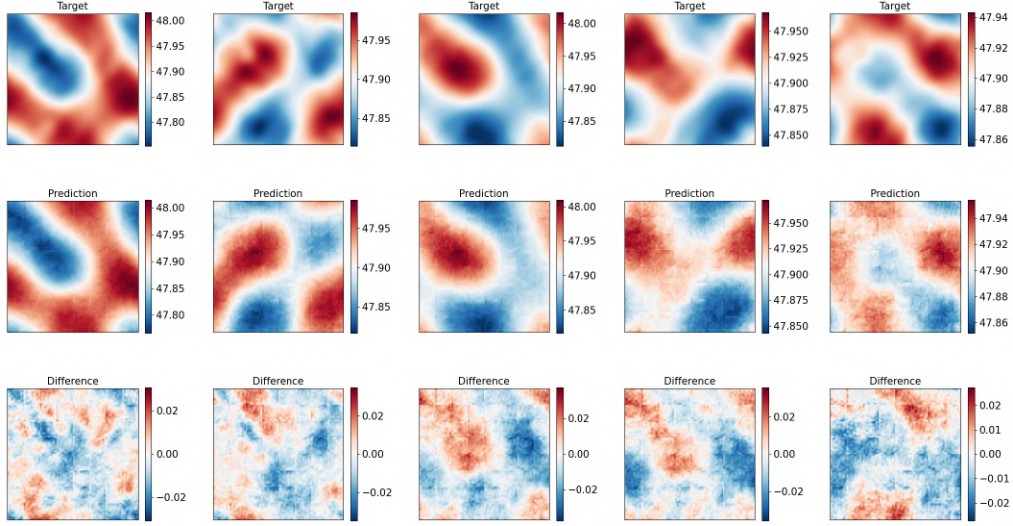

(a) A sequence of 5 output steps for PDEBench-Comp-HighVis dataset. The channel plotted is the pressure field in equation equation 12.

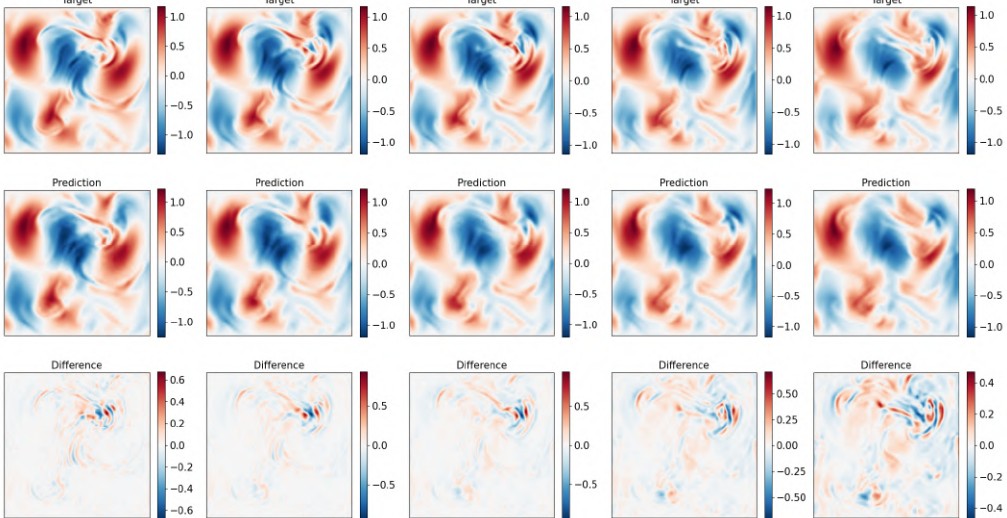

(b) A sequence of 5 output steps for PDEArena-Incomp dataset. The channel plotted is the x-velocity in equation equation 9.

Figure 12: **Example outputs for the VICON model.** (a) The pressure field of PDEBench-Comp-HighVis dataset, and (b) the x-velocity field of PDEArena-Incomp dataset. Each column represents a different timestep.

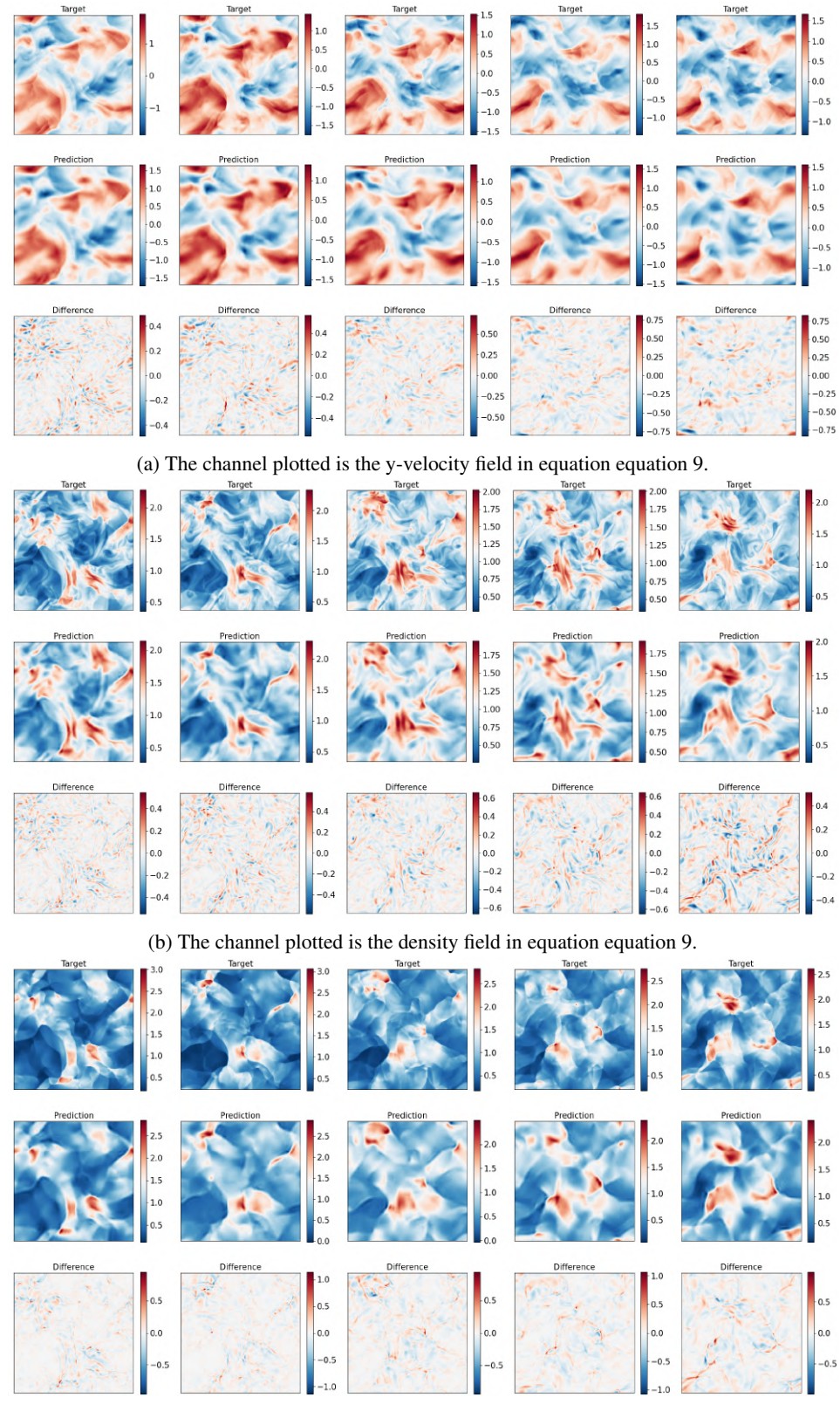

(a) The channel plotted is the y-velocity field in equation equation 9.

(b) The channel plotted is the density field in equation equation 9.

(c) The channel plotted is the pressure field in equation equation 9.

Figure 13: **More example outputs for the VICON model.** Showing 5 output steps for the PDEBench-Comp-LowVis dataset as governed by equation equation 9: (a) y-velocity, (b) density, and (c) pressure fields. Each column represents a different timestep.

