# OpenReview forum: "VICON: Vision In-Context Operator Networks for Multi-Physics Fluid Dynamics"
_NeurIPS.cc/2025/Conference — Submitted to NeurIPS 2025_

### Official Review · Reviewer_MQB3 · 2025-06-25

**Clarity:** 4
**Significance:** 3
**Originality:** 3
**Rating:** 5
**Confidence:** 3

**Summary:**

This paper presents VICON, a vision-transformer-based in-context operator network for multi-physics PDE prediction. The motivation is clear and well-justified: addressing the scalability issues of ICON in high-dimensional settings. The contributions are also clearly articulated, including an efficient patch-wise architecture and robust performance across diverse fluid systems. Overall, this is a well-structured and solid piece of work with strong motivation and clear contributions.

**Questions:**

1. *I have some concerns about the prediction stride.* While the flexible rollout design is a nice feature, it appears that in most cases, the best performance is still achieved when the stride is one. This is somewhat counter-intuitive: in traditional autoregressive training, a longer stride generally reduces error accumulation and improves performance for the same prediction horizon, since fewer steps are rolled out [1]. However, in Figure 5 of the appendix, “max step 7” seems to perform notably worse than “single step,” especially in the PDEBench datasets. Could the authors clarify why this occurs and whether it suggests a limitation in how VICON generalizes across stride lengths?
2. *I’m interested in how this framework could generalize beyond forward temporal prediction and 2D spatial domains.* While the authors mention in the limitations that extending to 3D or broader PDE tasks may face computational or domain expertise constraints, it would be helpful to have a more concrete estimate of how computational requirements might scale in 3D. Additionally, what specific forms of “specialized domain expertise” would be needed—for example, are we talking about mesh generation, physical modeling, or numerical solver integration? A more detailed discussion would help evaluate the feasibility of future extensions.

[1] Bi, Kaifeng, Lingxi Xie, Hengheng Zhang, Xin Chen, Xiaotao Gu, and Qi Tian. "Accurate medium-range global weather forecasting with 3D neural networks." Nature 619, no. 7970 (2023): 533-538.

**Ethical Concerns:**

["NO or VERY MINOR ethics concerns only"]

**Final Justification:**

The paper successfully extends the ICON framework to high-dimensional spatial domains by leveraging vision transformer architectures, significantly improving scalability while retaining the few-shot, in-context learning capability. I think the overall design and implementation are relatively simple and elegant, making the method easy to adopt and reproduce. Nevertheless, it seems still to be some gap for this framework to generalize beyond forward temporal prediction and 2D spatial domains, which makes the paper fall below the strong accept bar. Therefore, I vote for acceptance.

**Limitations:**

Yes, the authors thoroughly discussed the limitations of their work.

**Quality:**

4

**Strengths And Weaknesses:**

Strength:

1. The paper successfully extends the ICON framework to high-dimensional spatial domains by leveraging vision transformer architectures, significantly improving scalability while retaining the few-shot, in-context learning capability.

2. The method addresses practical challenges such as variable timestep rollout and imperfect temporal measurements, making it valuable for real-world applications.

3. The overall design and implementation are relatively simple and elegant, making the method easy to adopt and reproduce.

Weaknesses:

1. While flexible rollout is a core feature of VICON, in practice the best performance is often achieved with stride-1 predictions. This raises questions about how well the model truly benefits from variable stride training and whether longer-stride generalization is as effective as claimed. A deeper analysis of this behavior would be helpful.

2. The current study focuses primarily on autoregressive forecasting tasks for time-dependent PDEs, which somewhat narrows the broader applicability of the original ICON framework (e.g., to inverse problems or steady-state tasks). Extending VICON to support more diverse problem types would further enhance its utility.

---

> ### Author Rebuttal · Authors · 2025-07-29
>
> Thank you for recognizing the significance and clarity of VICON. We appreciate your recognition and here are the detailed response:
>
> > While flexible rollout is a core feature of VICON, in practice the best performance is often achieved with stride-1 predictions. This raises questions about how well the model truly benefits from variable stride training and whether longer-stride generalization is as effective as claimed. A deeper analysis of this behavior would be helpful.
>
> **Response:**
> We discuss this in Section 5.2, lines 270-275: PDEBench-Comp-LowVis and PDEBench-Comp-HighVis record data with larger timestep sizes, making the learning of larger stride predictions more challenging; also the fewer number of total rollout steps (even with stride \= 1\) cancels out the benefits of fewer number of rollouts when using multi-stride. On the other hand, PDEArena-Incomp records data with smaller timesteps, hence learning both single- and multi-stride prediction is not as hard; under this case, multi-stride reduces the total rollout steps and error accumulation.
>
> We will refine the writing for lines 270-275 to be more concrete and precise.
>
> > The current study focuses primarily on autoregressive forecasting tasks for time-dependent PDEs, which somewhat narrows the broader applicability of the original ICON framework (e.g., to inverse problems or steady-state tasks). Extending VICON to support more diverse problem types would further enhance its utility.
>
> **Response:**
> We thank you for suggesting additional task types. However, this kind of study on task variety has been extensively discussed in the original ICON paper \[1\]. As this work's core contribution is extending in-context neural operator learning to higher-dimensional spatial domains, we focus specifically on this advancement. We will add a comment in the revised version referring interested readers to \[1\] for broader task applicability discussions.
>
> > I have some concerns about the prediction stride. While the flexible rollout design is a nice feature, it appears that in most cases, the best performance is still achieved when the stride is one. This is somewhat counter-intuitive: in traditional autoregressive training, a longer stride generally reduces error accumulation and improves performance for the same prediction horizon, since fewer steps are rolled out [1]. However, in Figure 5 of the appendix, "max step 7" seems to perform notably worse than "single step," especially in the PDEBench datasets. Could the authors clarify why this occurs and whether it suggests a limitation in how VICON generalizes across stride lengths?
>
> **Response:**
> This is mainly because Δt \= 6, 7 are out-of-distribution and never seen during training (maximum stride during training \= 5, line 190). This demonstrates that VICON can generalize to some degree to out-of-distribution scenarios, as it can still extract reasonable dynamics from Δt \= 6, 7, albeit with some accuracy loss.
>
> However, it's worth noting that with larger timestrides, the dynamics will become more chaotic and harder to learn. How to choose a proper s\_max during training is currently a hyperparameter tuning in our framework. We can add this as a potential improvement for future work. Thank you again for mentioning this.
>
> > I'm interested in how this framework could generalize beyond forward temporal prediction and 2D spatial domains. While the authors mention in the limitations that extending to 3D or broader PDE tasks may face computational or domain expertise constraints, it would be helpful to have a more concrete estimate of how computational requirements might scale in 3D. Additionally, what specific forms of "specialized domain expertise" would be needed—for example, are we talking about mesh generation, physical modeling, or numerical solver integration? A more detailed discussion would help evaluate the feasibility of future extensions.
>
> **Response:**
> We will incorporate your suggested examples of expertise requirements, including mesh generation, physical modeling, and numerical solver integration.
>
> For computational scaling to 3D, we have considered two potential remedies (also suggested by R3): (1) using VAE-like techniques to compress the entire field into latent state vectors, and (2) using crop-like methods to learn the evolution of fixed-width windows (similar to CNNs). These approaches could potentially be combined to address the cubic scaling challenge in 3D domains.
>
> We will add these concrete possibilities and expertise requirements to our future work discussions.
>
> \[1\] In-context operator learning with data prompts for differential equation problems.

---

> > ### Comment · Reviewer_MQB3 · 2025-08-07
> >
> > Thank you for your response. The authors have addressed my concerns to a certain extent. Although the experimental section of the work is not perfect, the overall idea is novel and interesting. Therefore, I will maintain my accept recommendation.

---

### Official Review · Reviewer_38Wo · 2025-06-30

**Clarity:** 3
**Significance:** 2
**Originality:** 2
**Rating:** 4
**Confidence:** 4

**Summary:**

This paper introduces vision in-context operator networks (VICON), a new operator learning model that is designed to predict the dynamics of complex physical systems, with a specific focus on fluid dynamics. The paper builds upon the in-context operator network (ICON) framework but overcomes the limitation of computational cost in the ICON methodology which treats each spatial point as a token. It achieves this by integrating a vision transformer architecture, which would process the 2D spatial domain through patch-wise operations. The paper highlights strong performance when compared against the baselines of DPOT and MPP across 3 distinct benchmarks. In particular, as a consequence of its formulation, VICON excels in handling long-term predictions and handling of imperfect or irregular data.

**Questions:**

- Did you compare the performance of your model against more recent state-of-the-art foundation models like [Poseidon](https://arxiv.org/abs/2405.19101)?

- Have you explored in-context generalization of the architecture to different resolutions? Similar to time-step generalization, would this also prevail in space? In this case, it would involve rethinking the choice of learnable PEs

- Have you tried adding the stride in time as an additional conditioning variable? This way all the COND-QOI pairs may not require the use of the same stride within the same context

- I noticed in the results for the PDEBench-Comp-HighVis dataset, specifically in the error/difference maps shown in Figure 9 and Figure 12a, there appear to be block-like artifacts. This pattern seems to align with the model's 16x16 patch structure. I was curious as to why they're particularly visible in this case

- As I can see the number of tokens in an in-context learning framework would grow with increase in spatial resolution and in-context pairs. I know the authors have acknowledged this shortcoming. However, I was curious if you see a route to mitigate this cost especially when moving to higher resolution / complex problems / 3D?

**Ethical Concerns:**

["NO or VERY MINOR ethics concerns only"]

**Final Justification:**

I've chosen to provide a weak accept. The work is a technically interesting adaptation of the ICON framework to 2D, and the model's robustness to missing data is a notable strength.

This decision update comes after the discussion period. While the authors were highly engaged and confident in their defense, their rebuttals did not resolve scientific concerns that I'd raised (lack of testing on different resolutions / restrictive set of datasets). Further, I'm not entirely convinced about the efficiency advantage presented when the baselines compared against are only added in the sequence length. My rating has been raised based on the potential of the underlying framework rather than the conclusiveness of the current evidence.

**Limitations:**

The authors have been clear about the shortcomings of the model and outlined scope for future work such as 3D models, irregular grid, more general design beyond channel based masking. Maybe one downside of not having an additional spatial query that the model accepts is that the current methodology isn't amenable to a physics informed loss.

**Quality:**

3

**Strengths And Weaknesses:**

Strengths:

- The paper is well motivated and effectively highlights computational cost as a shortcoming of existing in-context operator networks (ICONs) as a result of point-wise tokenization. The proposed solution of using a patch-wise encoding is a logical approach to mitigate this cost
- The flexible time-step choice as a consequence of the in-context formulation is an elegant outcome. Further, the discussion on the robustness using the "imperfect" measurement study is an interesting demonstration which isn't often discussed for neural PDE surrogates. The innate ability to generalize across different rollout stride sizes using the same model is a particularly compelling result when compared against baselines.
- The reported improvements in computational efficiency of using a smaller number of parameters and faster inference when compared against baselines is a practical advantage.

Weaknesses:

- I think the experiments being performed aren't the most complete. While I commend the authors on building a foundation model across multiple fluid benchmarks by demonstrating the improvement obtained using joint training / individual training, I would like to highlight two shortcomings: (i) Firstly all the data processed by the model seems to live on a 128 x 128 grid. I would like to see the performance when including a similar fluid mechanics benchmark (perhaps a candidate could be the Rayleigh Benard / Rayleigh Taylor / Shear Flow benchmark from TheWell if possible), (ii) While already mentioned as a shortcoming, would it be possible to include a dataset that isn't NS based (perhaps diffusion-reaction).
- Further, it seems like the authors have only considered foundation models as a baseline of comparison. It would be helpful to see a comparison of the results obtained against other standard operator learning choices: FNO family and other vision transformer choices (ViTO, CViT). While these are bespoke models, they still provide a good reference of the cost of performance degradation at the cost of multi-physics capabilities.

---

> ### Author Rebuttal · Authors · 2025-07-29
>
> We thank you for your comments, here are the detailed responses:
>
> > I think the experiments being performed aren't the most complete. While I commend the authors on building a foundation model across multiple fluid benchmarks by demonstrating the improvement obtained using joint training / individual training, I would like to highlight two shortcomings: (i) Firstly all the data processed by the model seems to live on a 128 x 128 grid. I would like to see the performance when including a similar fluid mechanics benchmark (perhaps a candidate could be the Rayleigh Benard / Rayleigh Taylor / Shear Flow benchmark from TheWell if possible), (ii) While already mentioned as a shortcoming, would it be possible to include a dataset that isn't NS based (perhaps diffusion-reaction).
>
> **Response:**
> (i) Downsampling to 128×128 is common practice in prior works and ensures consistency with baselines for fair comparison. Methods like CNN+Transformer (e.g., SWIN\[1\]) may improve VICON for generalization to different resolutions. However, these side contributions would blur the **core advancement of this work: the very first extension of in-context neural operators to higher dimensions**. We therefore respectfully believe these extensions should be left as future work.
>
> (ii) Thank you for this suggestion. However, we respectfully believe that adding additional benchmarks would provide marginal value, as it would not change our core observations or conclusions.
>
> Our current experimental scope already covers diverse flow regimes as shown in the benchmark characteristics table below (from incompressible to compressible, divergence-free or not, dealing with discontinuities or not, from low Re to high Re), where our method demonstrates superior performance and unique advantages (e.g., handling imperfect measurement systems, a common challenge in real-world applications). We believe this establishes a solid foundation for our contributions, while recognizing that future work could explore additional PDE families.
>
> | Benchmark | Flow Type | Pressure-Velocity Coupling | Velocity Constraint | Discontinuity | Turbulence Treatment | Numerical Method |
> |-----------|-----------|---------------------------|--------------------|---------------|---------------------|------------------|
> | PDEArena-Incomp | Incompressible | Decoupled | Divergence-free | NA | NA | Fractional step method |
> | PDEBench-LowVis | Compressible | EOS | NA | Yes | High grid resolution or turbulence model | High resolution explicit |
> | PDEBench-HighVis | Compressible | EOS | NA | Yes | NA | Low resolution explicit |
>
> > Further, it seems like the authors have only considered foundation models as a baseline of comparison. It would be helpful to see a comparison of the results obtained against other standard operator learning choices: FNO family and other vision transformer choices (ViTO, CViT). While these are bespoke models, they still provide a good reference of the cost of performance degradation at the cost of multi-physics capabilities.
>
> **Response:**
> Thank you for suggesting single-physics baselines. However, we respectfully believe that **single-physics models and multi-physics models represent fundamentally different problem setups**. Providing such comparisons would potentially confuse readers regarding the core problem targeted: **in-context learning for multi-physics simulation without the need for fine-tuning**. This capability is **not achievable by single-physics models** unless with modifications (such as **appending attention layers to FNO as in DPOT, which we have already compared against**).
>
> > Did you compare the performance of your model against more recent state-of-the-art foundation models like Poseidon?
>
> **Response:**
> We have analyzed Poseidon and found it not a fair comparison to both VICON and seq2seq models like MPP/DPOT. Reasons follow:
>
> Poseidon **does not take any historical (sequence or pairs) information**, but performs autoregressive prediction **using only one initial frame**. This introduces **unresolvable ambiguity given the same initial condition but different dynamics**. For instance, in the PDEArena-Incomp dataset, different trajectories can share the same initial condition but contain **different working conditions: ie, randomized body forces.** Poseidon would **produce identical rollouts for all of them**, limiting its accuracy to at most one trajectory. To make it accurate, Poseidon then must be finetuned for every different (PDE, working condition) combinations, **this extensive finetuning requirement makes comparisons unfair even for seq2seq models like MPP/DPOT, let alone VICON.**
>
> > Have you explored in-context generalization of the architecture to different resolutions? Similar to time-step generalization, would this also prevail in space? In this case, it would involve rethinking the choice of learnable PEs
>
> **Response:**
> This echoes our previous response regarding architectural extensions. We acknowledge that such extension could be achieved using methods like SWIN\[1\], but **this is orthogonal to our core contribution.** We will add it to our future work discussions.
>
> > Have you tried adding the stride in time as an additional conditioning variable? This way all the COND-QOI pairs may not require the use of the same stride within the same context
>
> **Response:**
> **Explicitly encoding contextual variables is not scalable**—the environmental variables or working conditions of PDEs are **not limited to Δt**. For instance, PDEArena-Incomp also contains **randomized body forces**. As we incorporate more systems, each may have their different variables or conditions. It is **not feasible to exhaust all of them**.
>
> > I noticed in the results for the PDEBench-Comp-HighVis dataset, specifically in the error/difference maps shown in Figure 9 and Figure 12a, there appear to be block-like artifacts. This pattern seems to align with the model's 16x16 patch structure. I was curious as to why they're particularly visible in this case
>
> **Response:**
> These aliasing-like errors are common in patch-based methods, as observed also in MPP & DPOT, as there is no enforcement or alignment on the boundaries between patches. There are remedies such as using overlapping patches or moving patch positions during rollout—these are interesting engineering techniques, but we believe they are orthogonal to our core contributions. We would like to include such improvements as future work.
>
> > As I can see the number of tokens in an in-context learning framework would grow with increase in spatial resolution and in-context pairs. I know the authors have acknowledged this shortcoming. However, I was curious if you see a route to mitigate this cost especially when moving to higher resolution / complex problems / 3D?
>
> **Response:**
> Indeed, we have considered two remedies: (1) using **VAE-like techniques** to compress the entire field into latent state vectors, and (2) using **crop-like methods** to learn the evolution of fixed-width windows (similar to CNNs). These two approaches could also potentially be combined. We will add these concrete possibilities (also suggested by R4) to our future work discussions. Thank you for highlighting this important scaling challenge.
>
> \[1\] Swin Transformer: Hierarchical Vision Transformer using Shifted Windows.

---

> > ### Comment · Reviewer_38Wo · 2025-08-05
> > **Post-Rebuttal Discussion**
> >
> > Thank you for addressing the comments I had raised, and providing clarifying details. However, I still have a few questions I hope you can address.
> >
> > > These aliasing-like errors are common in patch-based methods, as observed also in MPP & DPOT, as there is no enforcement or alignment on the boundaries between patches. There are remedies such as using overlapping patches or moving patch positions during rollout—these are interesting engineering techniques, but we believe they are orthogonal to our core contributions. We would like to include such improvements as future work.
> >
> > I see. Thank you for clarifying this. I'm just a little surprised to see that it particularly shows up for this case while the other visualizations don't seem to have this checker boarding artifacts. Is there something particular to the grid structure that make them particularly appear in this case?
> >
> > > Downsampling to 128×128 is common practice in prior works and ensures consistency with baselines for fair comparison. Methods like CNN+Transformer (e.g., SWIN[1]) may improve VICON for generalization to different resolutions
> >
> > Thank you for clarifying this. Maybe one thing that come to mind on this front: Looking to confirm that the baseline comparisons also trained / evaluated with this resolution of inputs? Do you see any impact of the performance degradation in this case (if we choose to use the native 512 x 512 for the baselines). I would think that this should be significant if the flow contains a lot of fine scale features

---

> > > ### Author Response · Authors · 2025-08-06
> > > **Response to R3 Rebuttal**
> > >
> > > Thank you for the detailed discussion. We are glad to have resolved many of your concerns. Here are detailed responses for the remaining ones:
> > >
> > > **1\. Checkerboard artifacts in PDEBench-HighVis:**
> > > Following your suggestion, we checked these effects in the dataset and identified two contributing factors through detailed analysis:
> > >
> > > **Factor 1: Drastic field variation across trajectory time horizon.** High-viscosity flows rapidly dissipate kinetic energy, causing velocity fields to become nearly stationary after \~10 frames. Our analysis of standard deviations across different time phases reveals dramatic differences:
> > >
> > > | Dataset | density |  |  | u |  |  | v |  |  | pres |  |  |
> > > | :---- | :---- | :---- | :---- | :---- | :---- | :---- | :---- | :---- | :---- | :---- | :---- | :---- |
> > > |  | all | 1-10 | \>10 | all | 1-10 | \>10 | all | 1-10 | \>10 | all | 1-10 | \>10 |
> > > | PDEBench-HighVis | 2.656768 | 2.664458 | 2.649765 | **0.358999** | **0.511327** | **0.091414** | **0.351149** | **0.506210** | **0.049467** | 14.254244 | 14.495486 | 14.016444 |
> > > | PDEBench-LowVis | 3.388061 | 3.500075 | 3.282917 | 0.989394 | 1.103530 | 0.872773 | 1.036151 | 1.164948 | 0.903241 | 29.687168 | 29.933180 | 29.437323 |
> > >
> > > The bolded velocity channels show PDEBench-HighVis has extreme variation between early and late phases, unlike PDEBench-LowVis with nearly consistent variation. **This dynamic difference makes the training harder.**
> > >
> > > **Factor 2: Training-inference normalization mismatch.** During training, context pairs are sampled uniformly across all frames (by bootstrapping, the expected std is \~0.36 for velocity). During inference, context pairs are fixed from initial frames only (std \~0.51). **This normalization mismatch exacerbates prediction errors in the already-challenging problem.**
> > >
> > > Thank you for your detailed observation. **We will add this analysis to our manuscript and explore** using global statistics from multiple trajectories instead of single-trajectory instance normalization during deployment **(future work).**
> > >
> > > **2\. Resolution concern:**
> > > Baselines (MPP, DPOT) were not trained at 512×512 resolution. Most benchmark datasets simulate at 128×128 resolution (except PDEBench-LowVis which provides 512×512), so 128×128 is their native simulation resolution rather than a downsampled version. We use 128×128 consistently across all methods for fair comparison. Training at higher resolution would require increased model capacity to capture finer-scale dynamics, which remains an open challenge.
> > >
> > > Feel free to ask any more questions. If all of your concerns are resolved, we truly appreciate that you can re-consider the scoring.

---

> > > > ### Author Response · Authors · 2025-08-08
> > > > **Any follow up questions?**
> > > >
> > > > Dear reviewer 38Wo,
> > > >
> > > > We are glad to have resolved most of your confusions, and have replied to your concern on checkboard effects and resolution of the dataset (simulation). We are still waiting for your follow up comments on the last reply.
> > > >
> > > > **With less than 24 hrs of rebuttal period**, feel free to let us know if there is any more confusions we can discuss? if our rebuttal has resolved most of your comments, we would appreciate that you consider a rescoring.
> > > >
> > > > Best,

---

> ### Comment · Reviewer_38Wo · 2025-08-08
>
> Thank you for the clarification about the PDEBench-HighVis case. I hope you will add these details in the final version as well.
>
> Additionally, thank you for clarifying the training resolution being consistent at 128x128 for all problems.
>
> I still believe that the methodology would benefit from more extensive experiments across other resolutions, additional datasets especially since the it builds upon former in-context work (albeit extended to the 2D case now using patches). Further, as brought up in discussions with R2, I think the speed in training advantage / inference advantage would change when dealing with channel concatenation / perceiver aggregation (as performed in [CViT](https://openreview.net/forum?id=cRnCcuLvyr)) instead of current frames being concatenated along the sequence length. I don't entirely agree that only having models that concatenate frames along sequence length is a fair comparison.
>
> However, I think the in-context framework is elegant with advantages as demonstrated and would raise my rating to a borderline accept.

---

> > ### Author Response · Authors · 2025-08-09
> > **Thank you note and final discussion on channel concatenation**
> >
> > Dear reviewer 38Wo,
> >
> > Thank you so much for the for bringing up CViT to our attention, and agreeing to raising score to borderline accept. **We noticed you have not changed the original scoring, although the Mandatory Acknowledgement has been made; it could be a typo in your operation?**
> >
> > We have read the design of CViT and found it's combination of ViT as embedding + using attention to handle any location's query very interesting. This may eg resolve the checkboard effects R2 has mentioned before during chatting. We also found another related work that utilizes this method: [Li et al, 2023], https://arxiv.org/pdf/2205.13671. Although this is not multi-physics.
> >
> > As for the speed advantage would disappear when channel is longer. Indeed, this is a challenge we discussed in our limitation section. A very recent work on how to deal the the potential channel scaling issue (by inter-channel attention) when scaling up physics system types: [Holzschuh et al, 2025], https://arxiv.org/abs/2505.24717. It's too recent to be added as a baseline, but a good future work.
> >
> > **We will remember to discuss these two works as future work directions in our final revision.**
> >
> > **When you have time, can you take a look and fix the scoring on your original Official Review?**
> >
> > Best,

---

### Official Review · Reviewer_Looq · 2025-07-01

**Clarity:** 3
**Significance:** 3
**Originality:** 4
**Rating:** 4
**Confidence:** 3

**Summary:**

The authors introduce VICON, a vision transformer backbone that is integrated with in-context input-output pairs as part of the input sequence (sequence of patches from a 2D image grid) that can outperform SOTA models (DPOT and MPP) on spatiotemporal PDE emulation tasks. Specifically, they show the performance advantage on harder dynamical systems and systems with missing temporal samples (this mirrors some challenges in real-world), at a smaller training as well as inference cost. This motivates the use of VICON for PDE emulation tasks.

**Questions:**

In-context learning contributions: I feel some ablations would help put the results in better perspective
* As I understand, the vision transformer gets input-output pairs (u_t, u_(t+dt)) as part of the input sequence (and is concatenated) and hence the model has additional information about the inference task and can learn relationships purely through the above context (similar to LLMs). It seems like the first ablation is to show that a vision transformer without these pairs can fail in certain ways or have reduced performance. It would be good to see the same last-step rollout errors on in-distribution examples (from testing) and out-of-distribution examples (here, I would expect the vision transformer to fail steadily without prompting from the in-context examples). This strongly motivates VICON or in-fact any other model (MPP + ICON etc).
* It would be useful to demonstrate that VICON indeed takes advantage of the in-context examples. It is unclear now since the model architecture has also changed with the comparisons to DPOT and MPP which I assume were different architectures(?). A quick ablation is to produce u_(t+dt) from the in-distribution examples (say lower Re) as in-context examples when evaluating on high Re and demonstrate that the model performance reduces as it pays attention to the lower Re (the wrong) examples. You could make it harder by prompting with the identity map (u_t and u_t pairs) and see the performance degradation - these should be purely inference runs so hopefully this is a realistic experiment to run in the revision period.
* I am surprised at the computational expenses involved and would appreciate a discussion from the authors in the paper. VICON is cheaper to train and evaluate. I would expect that adding to the sequence would only blow up your computational costs (quadratically for the vision transformer). Maybe some quick back of the envelope calculations would be quite useful here to demonstrate why VICON is cheaper than DPOT/MPP. A related experiment from the first ablation suggested above is the following: the vanilla vision transformer would be cheaper to run than with extended sequence length. If you assume the same computational envelope, would it ever reach the performance of VICON (maybe through larger parameter count or through reduced patch size)? If so, then it motivates running larger DPOT models as well instead of just in-context examples.
* The joint training is interesting - is that true for other models (vision transformer included)? Here, I think the out-of-distribution error analysis would be particularly useful to see where the in-context examples help?
* Unless I misunderstood the performance of VICON, it seems like in-context examples are expensive and not scalable. For example, if we have several million pixels in the input frame, adding 10 other pairs would be prohibitive (100x inference costs). It might even be better to finetune in this case than rely on in-context (and offload cost to the training). To help understand this design space, it would be useful to show in-context vs finetuning performance on a computational envelope - this is mentioned in the introduction but there are no experiments to support that finetuning is worse and it requires 1000s of samples to reach the same accuracy as in-context learned models.

Other contributions:
* I may have misunderstood the temporal sampling: why is the MPP/DPOT framework not amenable to varying temporal step size? In principle, they could be trained on any sampling as well. Is this already done in the experiments? It seems like the simpler way to generalize to any time step size is to simply encode that information in the input rather than do any in-context training?

**Ethical Concerns:**

["NO or VERY MINOR ethics concerns only"]

**Final Justification:**

My clarifications have been addressed in the rebuttal phase and I now believe that this paper proposes a good (and novel) contribution. There were few results from the discussion that I hope the authors will include (non in-context ViT baselines, incorrect prompting results, costs discussions) in the paper which will strengthen it. The choice to compare with different architectures might pose difficulty in disentangling the in-context contributions but I appreciate that it is the norm to compare models against existing baselines.

**Limitations:**

Yes, some contextualization on scalability (see comments) above would be helpful to understand how the authors see the model being used in production.

**Paper Formatting Concerns:**

-

**Quality:**

2

**Strengths And Weaknesses:**

Strengths:
* Vision in-context models for SciML is a novel methodology with strong implications for future SciML models, especially for downstream task adaptation
* SOTA performance compared to MPP/DPOT with enhanced computational efficiency
* Methods are clear with good figures illustrating the model

Weaknesses:
* My main concern is that it is difficult to judge the effect of the in-context learning (the main contribution) from the experiments - some key ablations may be missing.
* [minor] The contributions are also not appropriately contextualized - there is the main contribution of the in-context vision model and some additional contributions with flexible temporal learning strategies that seems independent of the in-context model - again, some key ablations would help clarify these.

---

> ### Author Rebuttal · Authors · 2025-07-29
>
> Thank you for your comments. Below are our detailed responses:
>
> > Weaknesses: My main concern is that it is difficult to judge the effect of the in-context learning (the main contribution) from the experiments - some key ablations may be missing. [minor] The contributions are also not appropriately contextualized - there is the main contribution of the in-context vision model and some additional contributions with flexible temporal learning strategies that seems independent of the in-context model - again, some key ablations would help clarify these.
>
> **Response:**
> We will address all your specific ablation requests (from "questions" section) in detail below. Note that we have already included extensive ablations in the paper (Appendix D) and will provide additional experimental evidence in our responses. The claim that "key ablations may be missing" appears to overlook our comprehensive ablation studies on architectural designs, hyper-parameters choices, and robustness analysis.
>
> > Vision transformer without in-context pairs: It would be good to see the same last-step rollout errors on in-distribution examples (from testing) and out-of-distribution examples (here, I would expect the vision transformer to fail steadily without prompting from the in-context examples). This strongly motivates VICON or in-fact any other model.
>
> **Response:**
> Thank you for this suggestion. Your understanding of VICON’s design choice is precise, specifically for Markovian systems (lines 141-145). Once any initial frame is provided, we can predict the next frame without relying on previous history. This justifies forming condition-QoI pairs using only frames before/after specific timesteps. This design is a key difference from prior sequence-to-sequence baselines, which will fail when they **cannot form a complete sequence** (e.g., insufficient measurement length, missing frames) (empirical results in Section 5.3). This limitation gives our approach unique advantages beyond competitive performance metrics.
>
> Regarding out-of-distribution testing for VICON and baselines, using **unseen and large timestep strides** already constitutes an out-of-distribution test, as **Δt \= 6 or 7 are never observed during training** (Sec 5.2). Other baselines also cannot handle these large dts effectively—eg, they require frames at (0, 6, 12, ..., 54\) to form a sequence of length=10 with Δt \= 6, which exceeds the initial measurements (10 frames) and even the total length in PDEBench datasets (20 frames). This again highlights the flexibility of our context-as-pairs approach, which does not require such long consecutive measurements.
>
> > Demonstrating in-context example usage: It would be useful to demonstrate that VICON indeed takes advantage of the in-context examples. It is unclear now since the model architecture has also changed with the comparisons to DPOT and MPP which I assume were different architectures(?). A quick ablation is to produce u_(t+dt) from the in-distribution examples (say lower Re) as in-context examples when evaluating on high Re and demonstrate that the model performance reduces as it pays attention to the lower Re (the wrong) examples. You could make it harder by prompting with the identity map (u_t and u_t pairs) and see the performance degradation - these should be purely inference runs so hopefully this is a realistic experiment to run in the revision period.
>
> **Response:**
> Regarding the concern “the architectural detail differences, instead of the core in-context learning concept, is the factor of better performances". Architectural differences are inherent to any novel method, as it necessarily differs from prior works to some degree. To assure fairness, we have kept the number of parameters as consistent as possible (Table 11\) and maintained the same initial frame numbers.
>
> Regarding the concern “correct/incorrect example pairs significantly affect performance”, we agree and will add this ablation study by providing different levels of in-correct examples on the PDEArena-Incomp dataset (given the target is to 1-step prediction) in the revised version. A clear performance drop is observed with incorrect context pairs.
>
> | Experiment Setting | Average Rollout Error |
> | :---- | :---- |
> | Correct(with stride 1\) | 0.55 |
> | Mix context pairs with stride 1 and 2 | 0.88 |
> | Context pairs with stride 2 | 0.92 |
> | Random noise context pairs | 1.57 |
>
> > Computational cost analysis: I am surprised at the computational expenses involved and would appreciate a discussion from the authors in the paper. VICON is cheaper to train and evaluate. I would expect that adding to the sequence would only blow up your computational costs (quadratically for the vision transformer). Maybe some quick back of the envelope calculations would be quite useful here to demonstrate why VICON is cheaper than DPOT/MPP. A related experiment from the first ablation suggested above is the following: the vanilla vision transformer would be cheaper to run than with extended sequence length. If you assume the same computational envelope, would it ever reach the performance of VICON (maybe through larger parameter count or through reduced patch size)? If so, then it motivates running larger DPOT models as well instead of just in-context examples.
>
> **Response:**
> Regarding computational cost, the most important factor is the number of parameters where we have kept as consistent as possible (Table 11\) for fair comparison. The slower performance of DPOT may be due to additional FNO layers. MPP's axis-wise projects the whole domain into each 1D dimension but can still have high pixel counts there. There may also be engineering issues with baselines’ implementation, but addressing those is beyond the scope of this work.
>
> Regarding extending joint vs. single training to baselines, we respectfully believe the non-core ablations for other baselines is outside our scope.
>
> For the "in/out-distribution context effects on performance," this echoes a previous question—we kindly refer to Section 5.2 and the rollout experiments with Δt \= 6 and 7, which prove VICON can extract meaningful dynamics from unseen timesteps (out-of-distribution), though with some performance degradation.
>
> > In-context vs fine-tuning performance: Unless I misunderstood the performance of VICON, it seems like in-context examples are expensive and not scalable. For example, if we have several million pixels in the input frame, adding 10 other pairs would be prohibitive (100x inference costs). It might even be better to finetune in this case than rely on in-context (and offload cost to the training). To help understand this design space, it would be useful to show in-context vs finetuning performance on a computational envelope - this is mentioned in the introduction but there are no experiments to support that finetuning is worse and it requires 1000s of samples to reach the same accuracy as in-context learned models.
>
> **Response:**
> The "100x cost" claim appears to be a significant misunderstanding. **VICON incurs only 2²=4× computational cost compared to baselines** when all other settings (e.g., patch size) are identical, as it processes both condition and QoI, doubling the sequence length with quadratic attention complexity. Importantly, **this can be remedied at inference time** using KV caching, posing no significant practical troubles.
>
> Regarding whether in-context learning is worth it versus fine-tuning, this depends on the specific use case. **If the physical parameters (e.g., temperature or humidity) during deployment do not change**, then fine-tuning is applicable. However, when they **do change**, and you don't know when it will change, **adopting a fine-tuning approach online (with the correct timing to sense the change and re-fine-tuning) becomes troublesome**; while VICON does not need fine-tuning and provides superior flexibility. We do not claim VICON is superior in every scenario, but only its advantage for realistic deployment **where physical parameters can change and measurement systems can be imperfect**.
>
> > Temporal sampling strategy: I may have misunderstood the temporal sampling: why is the MPP/DPOT framework not amenable to varying temporal step size? In principle, they could be trained on any sampling as well. Is this already done in the experiments? It seems like the simpler way to generalize to any time step size is to simply encode that information in the input rather than do any in-context training?
>
> **Response:**
> This echoes our previous response discussing rollout with Δt \= 6 and 7\. For seq2seq models to incorporate large strides, they need **very long initial given frames**. For instance, they require frames at (0, 6, 12, ..., 54\) with Δt \= 6, which **exceed both the available initial measurements (10 frames) and even the total trajectory length in PDEBench datasets (20 frames)**. Even with Δt \= 2, the required frames are (0, 2, 4, ..., 18), **leaving merely 2 frames to predict** in PDEBench datasets, making the task trivial. This **fundamental limitation prevents seq2seq models** from effectively handling varying (especially large) timestep strides.

---

> > ### Comment · Reviewer_Looq · 2025-08-05
> >
> > Thank you to the authors for their response, I really appreciate it. I am still a little confused on few aspects so I would like to clarify a bit more in this discussion period.
> >
> > 1. Regarding the ViT baseline without in-context learning:
> >
> > The paper reads: "Notably, even
> > within the same dataset and fixed ∆t, different trajectories can exhibit different dynamics (e.g., due
> > to different Reynolds numbers Re). These variations are implicitly captured by the function pairs in
> > our framework." in the "Problem Setup" (l146- l148). My experiment request was to support this statement - vanilla ViT applied in a new Re setup that would fail without in-context prompting. I appreciate that the rebuttal period is short so if not possible, then maybe the problem setup can also emphasize that the paper tackles another (important) OOD setting with the time step change rather than the dynamics difference.
> >
> >
> > 2. Computational cost: I am still confused on the cost aspects.
> >
> > - "Regarding computational cost, the most important factor is the number of parameters " : I do not follow this. Arguably the most important aspect of the cost for a ViT is the spatiotemporal resolution. At any realistic problem scale (3D etc), the cost from self-attention is usually unmanageable due to the O(sequence_length**2) cost. Further, memory costs also will blow up primarily on the activation memory (and not the weights/number of parameters/optimizer states) for this reason.
> > - "The "100x cost" claim appears to be a significant misunderstanding. VICON incurs only 2²=4× computational cost compared to baselines": Again, I do not follow this still. Assuming my original sequence length is L (flattened 2D image), my cost is O(L**2). If I now add 10 more conditions at the input level for in-context learning, then the sequence length is now 10L and hence a 100x cost? Where does 4 come from?
> > - "Importantly, this can be remedied at inference time using KV caching, posing no significant practical troubles." Could you expand on this as well? I was under the impression KV cache is for autoregressive token by token prediction. As far as I understand, there is no temporal attention in VICON; further the full image is predicted in one shot by the ViT. Where does KV cache factor-in?
> >
> > 3. Temporal sampling:
> >
> > "For seq2seq models to incorporate large strides, they need very long initial given frames. For instance, they require frames at (0, 6, 12, ..., 54) with Δt = 6, which exceed both the available initial measurements (10 frames) and even the total trajectory length in PDEBench datasets (20 frames). "
> > Where does the number 10 frames come from? Isn't that a hyperparameter? I could have picked 2 frames? I'm basing this on my understanding of some of the high-end AI weather models like GraphCast (Google: https://www.science.org/doi/10.1126/science.adi2336) where they only use 2 frames. Now, the timestep can now be made variable by simply encoding it at the input?
> >
> > Thanks in advance, looking forward to your clarifications.

---

> ### Author Response · Authors · 2025-08-06
> **Response to R2 Rebuttal**
>
> Thank you for the detailed discussion. We are glad to have resolved many of your concerns. Here are detailed responses for the remaining ones:
>
> **1\. ViT baseline without in-context learning:**
> Thanks for your suggestion on adding experiments to show the importance of using in-context information. We trained a non-in-context seq2seq ViT with similar hyperparameter and architecture as VICON (162M parameters, 12 layers, with hidden dimension 1024 for attention, and 4096 for FFN, 10 initial frames given) on PDEArena-Incompress with Δt=1, then evaluated on both Δt=1 and Δt=2 (OOD):
>
> | Model | Stride | Last step (1e-2) | All average (1e-2) |
> | :---- | :---- | :---- | :---- |
> | ViT | 1 | 73.94 | 50.17 |
> | ViT | 2 | 95.03 | 63.15 |
>
> The significant performance degradation (95.03 vs 73.94) demonstrates that vanilla ViT cannot generalize to different Δt without in-context examples.
>
> **2\. Computational cost:**
> Thank you for providing your detailed analysis on complexity, which helped us understand that **your baseline definition differs from ours. You assume the baseline to be a model that takes a single input frame, but this can create fundamental ambiguity in multi-physics settings**. Consider 10 trajectories with identical initial conditions but different dynamics (different Re or body forces). A single-frame-as-input model would produce identical predictions regardless of the underlying different dynamics, making it accurate for at most one trajectory.
>
> Therefore, nearly all multi-physics learning baselines **(our definition) use seq2seq models that take multiple initial frames (in a consecutive sequence)** to resolve this ambiguity. Assume 10 initial frames given, the sequence lengths, respectively, for different methods are:
>
> - Single-frame model: 1 frame → p tokens
> - Seq2seq baselines: 10 frames → 10p tokens
> - VICON: 10 condition-qoi pairs → 20p tokens,
>
> where p \= patches per frame.
>
> The computational cost overhead then depends on the baseline definition:
>
> - Seq2seq vs. single-frame model: 10× sequence length → 100× cost
> - VICON vs. single-frame model: 20× sequence length → 400× cost
> - VICON vs. seq2seq (fair comparison in our work): 2× sequence length → 4× cost
>
> **3\. KV caching:**
> **Attention can operate over any token sequence, not just temporal sequences. VICON's attention operates over condition-qoi pairs** representing physical fields before/after certain timesteps. During inference, all context pairs except the final one pair (for latest prediction) are fixed, allowing their KV to be cached across rollout steps.
>
> **4\. Temporal sampling:**
> The 10-frame setup follows our **baselines (DPOT uses 10, MPP uses 16). We chose 10=min(10,16)** for the most challenging setting among baselines while maintain fairness.
>
> While **GraphCast** uses 2 frames, it **addresses a single-physics** weather prediction. For a **deterministic single-physics system, theoretically only 1 frame is needed** for rollout. However, **GraphCast uses 2 frames to estimate local stochasticity, as the weather system is a stochastic system** (also acknowledged as a limitation in their conclusions). We hypothesize **changing their problem to multi-physics settings will require more frames** to disambiguate both different dynamics and stochasticity.
>
> Feel free to ask any more questions. If all of your concerns are resolved, we truly appreciate that you can re-consider the scoring.

---

> > ### Comment · Reviewer_Looq · 2025-08-06
> >
> > Thank you for the clarifications. Some questions remain:
> >
> > Regarding 2 (computational cost):
> > I am still unclear. I understand that the Seq2Seq model will have 10 frames in the input. Going from the GraphCast example (which I assume fits into your Seq2Seq paradigm but with just 2 frames due to the single-physics system), the 10 frames are input as *channels* and not appended to the sequence. Assuming a ViT formulation, this is just embedded with patch embedding to some new dimension (embedding dimension). Hence, the cost is expensive at the patch embedding but unchanged through the architecture = O(L**2) attention still. However, VICON appends the 10 (x2 for the condition's output as well) frames to the *sequence* and the cost persists through the model (for how many every ViT blocks). Hence, attention is O((20L)**2) (I dropped the 2 in my previous estimate but it's not super important). So, it is a O(100)x more expensive to use VICON, theoretically. Could you clarify this?
> >
> > Regarding 4 (temporal sampling):
> > Yes, I understand the multiphysics task needs >1 frame. I was just wondering if the choice of 10 could have been 2/3/4 (something smaller) since it's more of a hyperparameter? And make it respect variable time steps by encoding dt at the input? But I also understand that you are respecting the baselines here so this is not going to factor in my score anymore.
> >
> > If the authors can clarify the computational cost a bit more, I would be happy to raise my score appropriately since the other questions were satisfactorily answered. I also appreciate the new baseline ViT and think it would be a good addition to the paper.

---

> > > ### Author Response · Authors · 2025-08-07
> > > **Further reply to reviewer Looq**
> > >
> > > We are happy to further clarify your remaining questions.
> > >
> > > **1\. Computational cost:**
> > >
> > > Thank you for the further explanation. Now we understand **our discrepancy stems from your assumption of channel concatenation instead of putting frames into sequence.**
> > >
> > > First, we note that **nearly all transformer-based seq2seq methods** for dynamics learning (beyond multi-physics learning) **put frames into the sequence length, rather than using channel concatenation**. Under this setting, our original O(4×) complexity compared to seq2seq is valid.
> > >
> > > Still, we can explain why channel concatenation isn't widely used. While theoretically feasible, channel concatenation creates significant complexity issues:
> > >
> > > 1. Each token/patch in current VICON consists of 8×8 pixels with 7 channels (448 degrees of freedom) embedded into dimension 1024\. If concatenating 10 frames along channels, each token would have 8×8×7×10=4480 degrees of freedom, requiring proportionally 10× increased hidden dimensions (e.g., 10240\) to prevent information loss.
> > >
> > > 2. The complexity of the transformer is **O(L²×D \+ L×D²)**, where D is the hidden dimension and L is sequence length. As we can see, **this complexity is symmetric in L and D**; hence even though **channel concatenation decreases L by 10-fold, the 10× increased hidden dimension D cancels out the benefits.**
> > >
> > > 3. Moreover, **modern transformer optimization** methods such as flash attention reduce memory usage linearly **in sequence length L**, but **no such optimization exists for hidden dimensions D**, making the channel approach more expensive than sequential processing.
> > >
> > > Finally, GraphCast is not a seq2seq model—it's a GNN-based approach without attention mechanisms on sequences/frames, so our previous analysis doesn't apply to it.
> > >
> > > **2\. Temporal sampling:**
> > > Yes, 10 frames is a hyperparameter matching our baselines. While explicit dt encoding \+ training with smaller dts is possible, it has two key limitations:
> > >
> > > a) **It will hurt** **extrapolation capability:** If a regression model is trained with inputs that all fall in \[0,1\], it would likely produce very wrong answers when evaluated at 1.5. Similarly, if we explicitly input dt and only train with small dts, the model's performance will likely drop significantly when fed with larger dt values explicitly in an input channel.
> > >
> > > Our current results in Section 5.2 show VICON can handle dt=6,7 (unseen during training), mainly because the flow fields forming the context pairs are still similar to those fields seen during training (e.g., those fields in pairs with dt=5).
> > >
> > > b) **Scalability:** Explicitly encoding contextual variables is not scalable—working conditions of PDEs extend beyond Δt. For instance, PDEArena-Incomp contains randomized body forces. As we incorporate more systems, each introduces different variables/conditions, making exhaustive explicit encoding infeasible. Our implicit approach using context pairs naturally handles this diversity.
> > >
> > > Thank you for your engagement. We believe these clarifications address the computational complexity concerns and appreciate your willingness to reconsider the score.

---

> > > > ### Comment · Reviewer_Looq · 2025-08-07
> > > >
> > > > Thank you very much for all the clarifications, I understand the costs now.
> > > >
> > > > I now suggest to the authors to include this discussion under the cost section - I am unsure if this naturally comes out of the paper in its current state. It's also important to acknowledge the 4x cost increase due to in-context examples so it contextualizes your inference numbers better - we would expect VICON to be slower than a non-in-context ViT but it's faster than MPP etc due to maybe other reasons. This also opens up questions about whether that 4x cost could be paid in finetuning (off-loaded costs) vs in-context and it would be nice to see a discussion on these tradeoffs as well.
> > > >
> > > > I raise my score to 4 now and hope the authors will include the new results and discussions in the revised version of the paper.

---

> > > > > ### Author Response · Authors · 2025-08-07
> > > > > **Thank you note and promise to incorperate revisions**
> > > > >
> > > > > Thanks reviewer Looq for the discussion.
> > > > >
> > > > > We look forward to this work and the potential impact it can bring to community. We will implement all revisions in the correct places.
> > > > >
> > > > > Best,

---

### Official Review · Reviewer_osck · 2025-07-03

**Clarity:** 3
**Significance:** 2
**Originality:** 2
**Rating:** 4
**Confidence:** 3

**Summary:**

The paper introduces Vision In‑Context Operator Networks (VICON), which augment the ICON framework by tokenizing 2D fields as vision‑transformer patches rather than individual points. This patch‑wise representation enables efficient in‑context learning of next‑step operators across multi‑physics fluid dynamics tasks.

**Questions:**

1. On line 134 the authors state that all training uses “the same PDE and consistent timestep size,” yet on line 88 they claim VICON is “a unified model for multiple PDEs.” Please clarify how VICON generalizes to different timesteps or entirely new PDEs or demonstrate zero‑shot transfer to a novel PDE or timestep at inference. Does the model require fine‑tuning or does it truly interpolate across unseen physics?
2. The PDEBench‑Comp experiments only extrapolate 10 frames, yet you advertise “Superior Performance on Long‑term Rollout Predictions.” If possible, provide results for significantly longer rollouts (e.g. 50–100 frames) to substantiate long‑term stability.
3.  In Figure 1(a), the top‑to‑bottom datasets are annotated with Δt = 1, 3, 2, which seems out of order. Please confirm whether the middle plot should be Δt = 2 (and bottom Δt = 3), or correct the figure caption.
4. The paper mentions both “patch” and “function” positional encodings but omits implementation details. Please describe how to compute and inject each encoding—e.g., learned vs. sinusoidal, per-patch vs. per-point—and where in the transformer block they’re added.
5. Lines 222–224 outline a “flexible-step” training strategy, but the description is hard to follow and Table 1 shows no accuracy gain over single‑step training. Please provide a step‑by‑step example of one flexible‑step training iteration and quantify its additional compute and explain why flexible‑step yields no apparent error reduction—what is its intended benefit?
6. Converting the field into fixed patches may introduce discretization error compared to point‑wise tokens. Please include a baseline with patch size = 1 (i.e., one token per grid point) to quantify any accuracy loss from patching.

**Ethical Concerns:**

["NO or VERY MINOR ethics concerns only"]

**Final Justification:**

The authors clarified their contributions and provided detailed elaborations, which mostly addressed my concerns about weaknesses; they also answered my questions regarding confusion and paper details. I think this is an appropriate work on exploring the integration of ViT's patch view into ICON models, only with some ambiguity and lack of evidence for specific claims, so I recommend Borderline accept.

**Limitations:**

yes

**Quality:**

3

**Strengths And Weaknesses:**

## Strengths
1. The paper innovatively combines vision-transformer patch tokenization with in-context operator networks, enabling efficient next-step prediction in multi-physics fluid simulations. The introduction of multiple timestep-sampling strategies further enhances flexibility across varying temporal resolutions.

2. A comprehensive suite of experiments—evaluating inference speed, rollout error, and robustness under dropped frames—convincingly demonstrates VICON’s effectiveness and clearly communicates its advantages over current baselines.

## Weaknesses

- Generality on irregular data: Treating fixed grid patches as tokens constrains applicability to regular domains; many real-world problems involve irregular meshes or unstructured data, where patch-wise operations may not extend directly.

- Training–inference mismatch: The reliance on teacher forcing during training assumes access to future ground truth, which is unavailable at deployment and differs from common operator-learning settings—this may artificially simplify learning and overestimate performance. A reduced dependence on ground truth (e.g., scheduled sampling) would better align with realistic use cases.

- Experimental diversity: All benchmarks stem from variations of the same underlying PDE; the model’s generalization to qualitatively different physics (e.g., multiphase flows, reacting flows) remains untested. Broader validation on disparate datasets would strengthen claims of versatility.

---

> ### Author Rebuttal · Authors · 2025-07-29
>
> We thank your suggestions and questions. The detailed responses are below:
>
> > Generality on irregular data: Treating fixed grid patches as tokens constrains applicability to regular domains; many real-world problems involve irregular meshes or unstructured data, where patch-wise operations may not extend directly.
>
> **Response:**
> This work represents the **first** extension of in-context neural operator learning to higher 2D dimensions—a significant computational and methodological achievement. The generalization to irregular grids, while valuable, represents a separate research direction orthogonal to our core contributions.
>
> **We have transparently discussed this limitation** in Section 6. As per the **review guidelines: "In general, authors should be rewarded rather than punished for being up front about the limitations of their work"**, we have comprehensively addressed scope limitations. We welcome the reviewer to identify any critical points missing from our analysis beyond those already acknowledged.
>
> > Training–inference mismatch: The reliance on teacher forcing during training assumes access to future ground truth, which is unavailable at deployment and differs from common operator-learning settings—this may artificially simplify learning and overestimate performance. A reduced dependence on ground truth (e.g., scheduled sampling) would better align with realistic use cases.
>
> **Response:**
> The reviewer's concern appears to be based on a fundamental misunderstanding of our method. **VICON does not use teacher forcing at any stage, nor does it require access to future ground truth during inference.**
>
> **Why VICON does not use teacher forcing:** During training, we form pairs of physical fields before/after a single, specific timestep—there are no "consecutive/autoregressive frames" beyond pairs in our input sequence. This eliminates the need for teacher forcing entirely, which is a core architectural advantage over sequence-to-sequence baselines.
>
> **No future ground truth needed during inference:** We form in-context pairs **exclusively using initially measured frames**, then perform autoregressive rollout by updating only the last pair's condition with our **latest prediction (never ground truth)**. This procedure (detailed in lines 213-215 and Appendix C.2) ensures zero access to future ground truth, making the concern about training-inference mismatch inapplicable to our method.
>
> > Experimental diversity: All benchmarks stem from variations of the same underlying PDE; the model's generalization to qualitatively different physics (e.g., multiphase flows, reacting flows) remains untested. Broader validation on disparate datasets would strengthen claims of versatility.
>
> **Response:**
> This comment reflects a misunderstanding of fluid dynamics complexity. **Fluid simulation encompasses a vast spectrum of physics that cannot be reduced to "variations of the same underlying PDE."** Our **3 benchmarks represent fundamentally distinct flow behaviors requiring entirely different numerical solvers**, as demonstrated in the table below:
>
> | Benchmark | Flow Type | Pressure-Velocity Coupling | Velocity Constraint | Discontinuity | Turbulence Treatment | Numerical Method |
> |-----------|-----------|---------------------------|--------------------|---------------|---------------------|------------------|
> | PDEArena-Incomp | Incompressible | Decoupled | Divergence-free | NA | NA | Fractional step method |
> | PDEBench-LowVis | Compressible | EOS | NA | Yes | High grid resolution or turbulence model | High resolution explicit |
> | PDEBench-HighVis | Compressible | EOS | NA | Yes | NA | Low resolution explicit |
>
> These fundamental differences necessitate completely different numerical solvers and represent distinct dynamics, even all of them are fluid simulations. **Our experimental scope comprehensively covers all of the typical flow regimes: compressible vs. incompressible, with/without discontinuities, laminar to turbulent flow regimes.** VICON's superior performance across these diverse scenarios, combined with unique advantages like handling imperfect measurements, demonstrates robust generalization within the typical flow regimes. Expanding to entirely different physics (e.g., reaction-diffusion) represents future work, not a limitation of current contributions.
>
> > On line 134 the authors state that all training uses "the same PDE and consistent timestep size," yet on line 88 they claim VICON is "a unified model for multiple PDEs." Please clarify how VICON generalizes to different timesteps or entirely new PDEs or demonstrate zero‑shot transfer to a novel PDE or timestep at inference. Does the model require fine‑tuning or does it truly interpolate across unseen physics?
>
> **Response:**
> To clarify: VICON indeed generalizes to different timesteps combined with different PDEs. **The confusion arises solely from unclear wording on line 134.**
>
> **We meant “... consistent within one sequence/row of the batch”**, ie, pairs must be formed from consistent PDE and timestep size to ensure a coherent example is shown. However, different combinations of dt and PDE are mixed **across different sequences/rows** within the training batch, for the model’s generalization ability to different combinations. We will revise the current line 134 in the revised version.
>
> > The PDEBench‑Comp experiments only extrapolate 10 frames, yet you advertise "Superior Performance on Long‑term Rollout Predictions." If possible, provide results for significantly longer rollouts (e.g. 50–100 frames) to substantiate long‑term stability.
>
> **Response:**
> We provide extensive long-term rollout results for the **PDEArena-Incomp dataset with 56 frames**. This dataset detail is presented in line 198, and the related results are visualized in Figure 2b, Figure 3c, Figure 4 (rightmost column), Figure 5c, Figure 7c, and Figure 8 (rightmost column).
>
> We indeed noticed a typo in line 536 ("...the initial frames span from time step 0 to 9, and we aim to predict the trajectory **up to time step 20**"), **this is only the case for two PDEBench datasets.** We will add the total steps for PDEArena-Incomp in the revised version.
>
> > In Figure 1(a), the top‑to‑bottom datasets are annotated with Δt = 1, 3, 2, which seems out of order. Please confirm whether the middle plot should be Δt = 2 (and bottom Δt = 3), or correct the figure caption.
>
> **Response:** We accidentally switched the legends for Δt \= 2 and Δt \= 3\. We will correct this in the revised manuscript.
>
> > The paper mentions both "patch" and "function" positional encodings but omits implementation details. Please describe how to compute and inject each encoding—e.g., learned vs. sinusoidal, per-patch vs. per-point—and where in the transformer block they're added.
>
> **Response:**
> Both encodings are learnable parameters (as stated in line 169). The positional encoding varies per patch (and is broadcast across points within each patch), while the functional encoding varies per condition/QoI in each pair (and is broadcast across patches and points of that condition/QoI). The specific dimensionalities are detailed in lines 169-173 (and in our code in supplementary material). We will enhance this description detail in the revision.
>
> > Lines 222–224 outline a "flexible-step" training strategy, but the description is hard to follow and Table 1 shows no accuracy gain over single‑step training. Please provide a step‑by‑step example of one flexible‑step training iteration and quantify its additional compute and explain why flexible‑step yields no apparent error reduction—what is its intended benefit?
>
> **Response:**
>
> We kindly note that **we do not claim  "flexible training" but write "flexible rollout"** (lines 222-224).
>
> Regarding “why no apparent error reduction in single-step prediction”, and what is the benefit of **"flexible rollout"**. Learning predictions under different timestep sizes is inherently a broader and more challenging task than that under one stepsize. However, you have the flexibility to reduce the number of autoregressive steps (where each step incurs a prediction error, hence in chain, error accumulation). e.g., if max stride is 5, we can predict 11→16→21→...→51→56 (9 autoregressive steps) instead of 11→12→13→...→55→56 (45 autoregressive steps), significantly decreases the number of error accumulation as 1/5. This empirically explains our superior long-term predictions for longer sequences, as detailed in Sections 5.2-5.3 and exemplified by the rollout schedules in Table 5\.
>
> > Converting the field into fixed patches may introduce discretization error compared to point‑wise tokens. Please include a baseline with patch size = 1 (i.e., one token per grid point) to quantify any accuracy loss from patching.
>
> **Response:**
> **First regarding the discretization error,** each patch has 16\*16 pixels with at most 4 meaningful channels (16\*16\*4=1024), since we convert each patch into a feature vector of dimension 1024, theoretically this should be enough expressibility and hence no loss of information.
>
> Regarding the recommended ablation study of patch size \=1; We kindly note that we already include an ablation study on patch resolutions in Appendix D.1. The minimum patch size we can afford is 8×8, and the latent dimension must be reduced to fit within memory constraints, which leads to performance downgrade. Any resolution lower than 8×8 exceeds our computational memory constraints (lines 566-567).
>
> This computational complexity (of not using patch) is exactly the motivation of this work (lines 55-59). We will reiterate this motivation more clearly in the results section (lines 246-247, 264\) to better remind the readers.

---

> ### Comment · Reviewer_osck · 2025-08-06
>
> Many thanks to the authors for the detailed elaborations, which mostly addressed my concerns. However, I'm not confirmed about the teacher forcing problem. As far as I understand, the training procedure is to reconstruct the output function given input-output pair, not involving predicting further frames. So the task is to align with part of the model input (ultimately from the dataset, not generated by the model, which can be seen as ground truth) without performing future rollout, nor minimizing the gap between prediction and unseen real field. To my knowledge, this is a kind of teacher forcing that causes training–inference mismatch. Could you please explain that?

---

> ### Author Response · Authors · 2025-08-06
> **Response to R1 Rebuttal**
>
> Dear reviewer, thank you for the clarification and continued discussion. Rather than debating teacher-forcing definitions (our understanding may be too strict), **let's address the core related question: does VICON suffer from training-inference mismatch, and how does this compare to seq2seq baselines?**
>
> 1. **Yes, VICON suffers from training-inference mismatch.** As discussed earlier, during rollout, the final pair (to make the latest prediction) uses the (autoregressively) predicted frame as condition, while other pairs remain unchanged from clean initial data. The **final pair's condition can suffer from accumulated error, causing training-inference misalginment.** This is usually referred to as **error accumulation** in our domain.
>
> 2. **Still, VICON suffers less than seq2seq baselines.** E.g., assuming 10 frames are given initially, after 10 autoregressive steps:
>
>    * **Seq2seq:** All frames in current window (10,11,...,19) contain accumulated errors
>    * **VICON:** Only 1/10 (the last) context pairs involves predicted data; 9/10 remain error-free from initial frames
>
> Both intuitive reasoning and empirical evidence demonstrate that VICON's architectural advantage significantly reduces error propagation in longer rollouts.
>
> Still, how to resolve error accumulation can be a parallel contribution to us. It may be resolved by injecting Gaussian noise \[1\] or using push-forward tricks \[2\]. We are glad to add these potential improvements to the future work section.
>
> Feel free to ask us anymore questions. If all of your concerns are resolved, we truly appreciate that you can re-consider the scoring.
>
> \[1\] https://arxiv.org/abs/2010.03409
> \[2\] https://arxiv.org/abs/2202.03376

---

> > ### Comment · Reviewer_osck · 2025-08-06
> >
> > I appreciate the authors directly facing the core problem, and I don't have further concerns. I hope the authors integrate the revisions we discussed above to make a polished updated version of the manuscript. My score will accordingly rise to borderline accept.

---

> > > ### Author Response · Authors · 2025-08-06
> > > **Thank you note and promise to integrate discussions**
> > >
> > > Thank you so much for your updates!
> > >
> > > Yes, we truely look forward to this work and the potential impact it can bring to community. We will implement all revisions in the correct places.
> > >
> > > Best,

---

### Decision · Program_Chairs · 2025-09-17

**Decision:**

Reject

**Comment:**

This work introduces a (block causal) vision transformer ViT into the ICON framework for in-context learning of "operators" (time-dependent PDEs), i.e., regression of solutions for new conditions, given a context set of conditions and outputs. The key difference wrt prior work "ICON" is in the tokenization of the problem, where instead of assigning a token to each data-point, "patches" of data serve as tokens.

The reviewers appreciated the thorough experiment, and robustness experiments conducted on (three) fluid dynamics benchmarks. However, a key claim of faster inference time has been pointed out by the reviewers to not be concordant with 4x longer sequence lengths/ cost wrt baseline seq2seq methods. Further there are concerns regarding under-exploration of alternative tokenization strategies, e.g., channel concatenation and perceiver aggregation. Also, the applications explored in the evaluations are limited to a single domain, and extensions to higher-dimensions and other datasets/problems. Finally, the change in tokenization and introduction of the vision transformer in the ICON framework is a relatively incremental development.

In view of the above, I am inclined to not recommend this work in its current form. I encourage the authors to further develop their work by incorporating the valuable reviewer feedback.